# Weisfeiler and Leman go sparse: Towards scalable higher-order graph embeddings

Christopher Morris[*]          Gaurav Rattan[†]          Petra Mutzel[‡]

## Abstract

Graph kernels based on the 1-dimensional Weisfeiler-Leman algorithm and corresponding neural architectures recently emerged as powerful tools for (supervised) learning with graphs. However, due to the purely local nature of the algorithms, they might miss essential patterns in the given data and can only handle binary relations. The $k$-dimensional Weisfeiler-Leman algorithm addresses this by considering $k$-tuples, defined over the set of vertices, and defines a suitable notion of adjacency between these vertex tuples. Hence, it accounts for the higher-order interactions between vertices. However, it does not scale and may suffer from overfitting when used in a machine learning setting. Hence, it remains an important open problem to design WL-based graph learning methods that are simultaneously expressive, scalable, and non-overfitting. Here, we propose *local* variants and corresponding neural architectures, which consider a subset of the original neighborhood, making them more scalable, and less prone to overfitting. The expressive power of (one of) our algorithms is strictly higher than the original algorithm, in terms of ability to distinguish non-isomorphic graphs. Our experimental study confirms that the local algorithms, both kernel and neural architectures, lead to vastly reduced computation times, and prevent overfitting. The kernel version establishes a new state-of-the-art for graph classification on a wide range of benchmark datasets, while the neural version shows promising performance on large-scale molecular regression tasks.

## 1   Introduction

Graph-structured data is ubiquitous across application domains ranging from chemo- and bioinformatics [10, 103] to image [101] and social network analysis [27]. To develop successful machine learning models in these domains, we need techniques that can exploit the rich information inherent in the graph structure, as well as the feature information contained within nodes and edges. In recent years, numerous approaches have been proposed for machine learning with graphs—most notably, approaches based on *graph kernels* [71] or using *graph neural networks* (GNNs) [19, 45, 47]. Here, graph kernels based on the 1-*dimensional Weisfeiler-Leman algorithm* (1-WL) [46, 111], and corresponding GNNs [83, 115] have recently advanced the state-of-the-art in supervised node and graph learning. Since the 1-WL operates via simple neighborhood aggregation, the purely local nature of these approaches can miss important patterns in the given data. Moreover, they are only applicable to binary structures, and therefore cannot deal with general $t$-ary structures, e.g., hypergraphs [124] or subgraphs, in a straight-forward way. A provably more powerful algorithm (for graph isomorphism testing) is the $k$-*dimensional Weisfeiler-Leman algorithm* ($k$-WL) [15, 46, 77]. The algorithm can capture more global, higher-order patterns by iteratively computing a coloring (or discrete labeling) for $k$-tuples, instead of single vertices, based on an appropriately defined notion of adjacency between two $k$-tuples. However, it fixes the cardinality of this neighborhood to $k \cdot n$, where $n$ denotes the number of

---

[*]CERC in Data Science for Real-Time Decision-Making, Polytechnique Montréal
[†]Department of Computer Science, RWTH Aachen University
[‡]Department of Computer Science, University of Bonn

vertices of a given graph. Hence, the running time of each iteration does not take the *sparsity* of a given graph into account. Further, new neural architectures [77, 78] that possess the same power as the $k$-WL in terms of separating non-isomorphic graphs suffer from the same drawbacks, i.e., they have to resort to dense matrix multiplications. Moreover, when used in a machine learning setting with real-world graphs, the $k$-WL may capture the isomorphism type, which is the complete structural information inherent in a graph, after only a couple of iterations, which may lead to overfitting, see [82], and the experimental section of the present work.

**Present work** To address this, we propose a *local* version of the $k$-WL, the *local $\delta$-$k$-dimensional Weisfeiler-Leman algorithm* ($\delta$-$k$-LWL), which considers a subset of the original neighborhood in each iteration. The cardinality of the *local neighborhood* depends on the sparsity of the graph, i.e., the degrees of the vertices of a given $k$-tuple. We theoretically analyze the strength of a variant of our local algorithm and prove that it is strictly more powerful in distinguishing non-isomorphic graphs compared to the $k$-WL. Moreover, we devise a hierarchy of pairs of non-isomorphic graphs that a variant of the $\delta$-$k$-LWL can separate while the $k$-WL cannot. On the neural side, we devise a higher-order graph neural network architecture, the $\delta$-$k$-LGNN, and show that it has the same expressive power as the $\delta$-$k$-LWL. Moreover, we connect it to recent advancements in learning theory for GNNs [41], which show that the $\delta$-$k$-LWL architecture has better generalization abilities compared to dense architectures based on the $k$-WL. See Figure 1 for an overview of the proposed algorithms.

Experimentally, we apply the discrete algorithms (or kernels) and the (local) neural architectures to supervised graph learning, and verify that both are several orders of magnitude faster than the global, discrete algorithms or dense, neural architectures, and prevent overfitting. The discrete algorithms establish a new state-of-the-art for graph classification on a wide range of small- and medium-scale classical datasets. The neural version shows promising performance on large-scale molecular regression tasks.

**Related work** In the following, we review related work from graph kernels and GNNs. We refer to Appendix A for an in-depth discussion of related work, as well as a discussion of theoretical results for the $k$-WL.

Historically, kernel methods—which implicitly or explicitly map graphs to elements of a Hilbert space—have been the dominant approach for supervised learning on graphs. Important early work in this area includes kernels based on random-walks [42, 60, 70], shortest paths [13], and kernels based on the 1-WL [100]. Morris et al. [82] devised a local, set-based variant of the $k$-WL. However, the approach is (provably) weaker than the tuple-based algorithm, and they do not prove convergence to the original algorithm. For a thorough survey of graph kernels, see [71]. Recently, graph neural networks (GNNs) [45, 97] emerged as an alternative to graph kernels. Notable instances of this architecture include, e.g., [33, 51, 105], and the spectral approaches proposed in, e.g., [14, 29, 64, 81]—all of which descend from early work in [65, 80, 102, 97]. A survey of recent advancements in GNN techniques can be found, e.g., in [19, 113, 125]. Recently, connections to Weisfeiler-Leman type algorithms have been shown [11, 24, 43, 44, 75, 77, 83, 115]. Specifically, the authors of [83, 115] showed that the expressive power of any possible GNN architecture is limited by the 1-WL in terms of distinguishing non-isomorphic graphs. Morris et al. [83] introduced $k$-*dimensional GNNs* ($k$-GNN) which rely on a message-passing scheme between subgraphs of cardinality $k$. Similar to [82], the paper employed a local, set-based (neural) variant of the $k$-WL, which is (provably) weaker than the variant considered here. Later, this was refined in [77] by introducing $k$-*order invariant graph networks* ($k$-IGN), based on Maron et al. [78], which are equivalent to the folklore variant of the $k$-WL [44, 46] in terms of distinguishing non-isomorphic graphs. However, $k$-IGN may not scale since they rely on dense linear algebra routines. Chen et al. [24] connect the theory of universal approximation of permutation-invariant functions and the graph isomorphism viewpoint and introduce a variation of the 2-WL, which is more powerful than the former. Our comprehensive treatment of higher-order, sparse, (graph) neural networks for arbitrary $k$ subsumes all of the algorithms and neural architectures mentioned above.

## 2   Preliminaries

We briefly describe the Weisfeiler-Leman algorithm and, along the way, introduce our notation, see Appendix B for expanded preliminaries. As usual, let $[n] = \{1, \ldots, n\} \subset \mathbb{N}$ for $n \geq 1$, and let $\{\!\{\ldots\}\!\}$ denote a multiset. We also assume elementary definitions from graph theory (such as

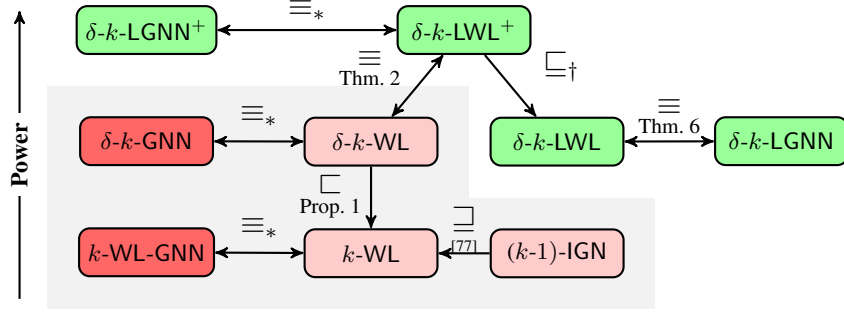

Figure 1: Overview of the power of proposed algorithms and neural architectures. The green and dark red nodes represent algorithms proposed in the present work. The grey region groups dense algorithms and neural architectures.

$*$—Follows directly from the proof of Theorem 6. $A \sqsubseteq B$ ($A \sqsubset B$, $A \equiv B$): algorithm $A$ is more powerful (strictly more powerful, equally powerful) than $B$, $\dagger$—Follows by definition, strictness open.

graphs, directed graphs, vertices, edges, neighbors, trees, and so on). The vertex and the edge set of a graph $G$ are denoted by $V(G)$ and $E(G)$ respectively. The *neighborhood* of $v$ in $V(G)$ is denoted by $\delta(v) = N(v) = \{u \in V(G) \mid (v, u) \in E(G)\}$. Moreover, its complement $\bar{\delta}(v) = \{u \in V(G) \mid (v, u) \notin E(G)\}$. We say that two graphs $G$ and $H$ are *isomorphic* ($G \simeq H$) if there exists an adjacency preserving bijection $\varphi \colon V(G) \to V(H)$, i.e., $(u, v)$ is in $E(G)$ if and only if $(\varphi(u), \varphi(v))$ is in $E(H)$, call $\varphi$ an *isomorphism* from $G$ to $H$. If the graphs have vertex/edges labels, the isomorphism is additionally required to match these labels. A *rooted tree* is a tree with a designated vertex called *root* in which the edges are directed in such a way that they point away from the root. Let $p$ be a vertex in a directed tree then we call its out-neighbors *children* with parent $p$. Given a $k$-tuple of vertices $\mathbf{v} = (v_1, \ldots, v_k)$, let $G[\mathbf{v}]$ denote the subgraph induced on the set $\{v_1, \ldots, v_k\}$, where, the vertex $v_i$ is labeled with $i$, for $i$ in $[k]$.

**Vertex refinement algorithms** For a fixed positive integer $k$, let $V(G)^k$ denote the set of $k$-tuples of vertices of $G$. A *coloring* of $V(G)^k$ is a mapping $C \colon V(G)^k \to \mathbb{N}$, i.e., we assign a number (or color) to every tuple in $V(G)^k$. The *initial coloring* $C_0$ of $V(G)^k$ is specified by the isomorphism types of the tuples, i.e., two tuples $\mathbf{v}$ and $\mathbf{w}$ in $V(G)^k$ get a common color iff the mapping $v_i \to w_i$ induces an isomorphism between the labeled subgraphs $G[\mathbf{v}]$ and $G[\mathbf{w}]$. A *color class* corresponding to a color $c$ is the set of all tuples colored $c$, i.e., the set $C^{-1}(c)$. For $j$ in $[k]$, let $\phi_j(\mathbf{v}, w)$ be the $k$-tuple obtained by replacing the $j^{\text{th}}$ component of $\mathbf{v}$ with the vertex $w$. That is, $\phi_j(\mathbf{v}, w) = (v_1, \ldots, v_{j-1}, w, v_{j+1}, \ldots, v_k)$. If $\mathbf{w} = \phi_j(\mathbf{v}, w)$ for some $w$ in $V(G)$, call $\mathbf{w}$ a *$j$-neighbor* of $\mathbf{v}$ (and vice-versa). The *neighborhood* of $\mathbf{v}$ is then defined as the set of all tuples $\mathbf{w}$ such that $\mathbf{w} = \phi_j(\mathbf{v}, w)$ for some $j$ in $[k]$ and $w$ in $V(G)$. The *refinement* of a coloring $C \colon V(G)^k \to \mathbb{N}$, denoted by $\widehat{C}$, is a coloring $\widehat{C} \colon V(G)^k \to \mathbb{N}$ defined as follows. For each $j$ in $[k]$, collect the colors of the $j$-neighbors of $\mathbf{v}$ as a multiset $S_j = \{\!\{C(\phi_j(\mathbf{v}, w)) \mid w \in V(G)\}\!\}$. Then, for a tuple $\mathbf{v}$, define $\widehat{C}(\mathbf{v}) = (C(\mathbf{v}), M(\mathbf{v}))$, where $M(\mathbf{v})$ is the $k$-tuple $(S_1, \ldots, S_k)$. For consistency, the strings $\widehat{C}(\mathbf{v})$ thus obtained are lexicographically sorted and renamed as integers. Observe that the new color $\widehat{C}(\mathbf{v})$ of $\mathbf{v}$ is solely dictated by the color histogram of its neighborhood and the previous color of $\mathbf{v}$. In general, a different mapping $M(\cdot)$ could be used, depending on the neighborhood information that we would like to aggregate.

**The $k$-dim. Weisfeiler-Leman** For $k \geq 2$, the $k$-WL computes a coloring $C_\infty \colon V(G)^k \to \mathbb{N}$ of a given graph $G$, as follows.[4] To begin with, the initial coloring $C_0$ is computed. Then, starting with $C_0$, successive refinements $C_{i+1} = \widehat{C_i}$ are computed until convergence. That is,

$$C_{i+1}(\mathbf{v}) = (C_i(\mathbf{v}), M_i(\mathbf{v})),$$

where

$$M_i(\mathbf{v}) = \big(\{\!\{C_i(\phi_1(\mathbf{v}, w)) \mid w \in V(G)\}\!\}, \ldots, \{\!\{C_i(\phi_k(\mathbf{v}, w)) \mid w \in V(G)\}\!\}\big). \qquad (1)$$

The successive refinement steps are also called *rounds* or *iterations*. Since the disjoint union of the color classes form a partition of $V(G)^k$, there must exist a finite $\ell \leq |V(G)|^k$ such that $C_\ell = \widehat{C_\ell}$. In

the end, the $k$-WL outputs $C_\ell$ as the *stable coloring* $C_\infty$. The $k$-WL *distinguishes* two graphs $G$ and $H$ if, upon running the $k$-WL on their disjoint union $G \dot\cup H$, there exists a color $c$ in $\mathbb{N}$ in the stable coloring such that the corresponding color class $S_c$ satisfies $|V(G)^k \cap S_c| \neq |V(H)^k \cap S_c|$, i.e., there exist an unequal number of $c$-colored tuples in $V(G)^k$ and $V(H)^k$. Hence, two graphs distinguished by the $k$-WL must be non-isomorphic. See Appendix C for its relation to the folklore $k$-WL.

**The $\delta$-$k$-dim. Weisfeiler-Leman** Let $\mathbf{w} = \phi_j(\mathbf{v}, w)$ be a $j$-neighbor of $\mathbf{v}$. Call $\mathbf{v}$ a *local $j$-neighbor* of $\mathbf{w}$ if $w$ is adjacent to the replaced vertex $v_j$. Otherwise, call $\mathbf{v}$ a *global $j$-neighbor* of $\mathbf{w}$. For tuples $\mathbf{v}$ and $\mathbf{w}$ in $V(G)^k$, let the function $\mathrm{adj}((\mathbf{v}, \mathbf{w}))$ evaluate to L or G, depending on whether $\mathbf{w}$ is a local or a global neighbor, respectively, of $\mathbf{v}$. The $\delta$-$k$-dimensional *Weisfeiler-Leman algorithm*, denoted by $\delta$-$k$-WL, is a variant of the classic $k$-WL which *differentiates* between the local and the global neighbors during neighborhood aggregation [76]. Formally, the $\delta$-$k$-WL algorithm refines a coloring $C_i$ (obtained after $i$ rounds) via the aggregation function

$$M_i^{\delta,\overline{\delta}}(\mathbf{v}) = \big( \{\!\!\{ (C_i(\phi_1(\mathbf{v}, w), \mathrm{adj}(\mathbf{v}, \phi_1(\mathbf{v}, w)))) \mid w \in V(G) \}\!\!\}, \ldots, \\ \{\!\!\{ (C_i(\phi_k(\mathbf{v}, w), \mathrm{adj}(\mathbf{v}, \phi_k(\mathbf{v}, w)))) \mid w \in V(G) \}\!\!\} \big), \tag{2}$$

instead of the $k$-WL aggregation specified by Equation (1). We define the 1-WL to be the $\delta$-1-WL, which is commonly known as color refinement or naive vertex classification.

**Comparison of $k$-WL variants** Let $A_1$ and $A_2$ denote two vertex refinement algorithms, we write $A_1 \sqsubseteq A_2$ if $A_1$ distinguishes between all non-isomorphic pairs $A_2$ does, and $A_1 \equiv A_2$ if both directions hold. The corresponding strict relation is denoted by $\sqsubset$. The following result shows that the $\delta$-$k$-WL is strictly more powerful than the $k$-WL for $k \geq 2$ (see Appendix C.1.1 for the proof).

**Proposition 1.** For $k \geq 2$, the following holds:

$$\delta\text{-}k\text{-WL} \sqsubset k\text{-WL}.$$

## 3 Local $\delta$-$k$-dimensional Weisfeiler-Leman algorithm

In this section, we define the new *local $\delta$-$k$-dimensional Weisfeiler-Leman algorithm* ($\delta$-$k$-LWL). This variant of the $\delta$-$k$-WL considers only local neighbors during the neighborhood aggregation process, and discards any information about the global neighbors. Formally, the $\delta$-$k$-LWL algorithm refines a coloring $C_i$ (obtained after $i$ rounds) via the aggregation function,

$$M_i^{\delta}(\mathbf{v}) = \big( \{\!\!\{ C_i(\phi_1(\mathbf{v}, w)) \mid w \in N(v_1) \}\!\!\}, \ldots, \{\!\!\{ C_i(\phi_k(\mathbf{v}, w)) \mid w \in N(v_k) \}\!\!\} \big), \tag{3}$$

instead of Equation (2), hence considering only the local $j$-neighbors of the tuple $\mathbf{v}$ in each iteration. The indicator function $\mathrm{adj}$ used in Equation (2) is trivially equal to L here, and is thus omitted. The coloring function for the $\delta$-$k$-LWL is then defined by

$$C_{i+1}^{k,\delta}(\mathbf{v}) = (C_i^{k,\delta}(\mathbf{v}), M_i^{\delta}(\mathbf{v})).$$

We also define the $\delta$-$k$-LWL$^+$, a minor variation of the $\delta$-$k$-LWL. Later, we will show that the $\delta$-$k$-LWL$^+$ is equivalent in power to the $\delta$-$k$-WL (Theorem 2). Formally, the $\delta$-$k$-LWL$^+$ algorithm refines a coloring $C_i$ (obtained after $i$ rounds) via the aggregation function,

$$M^{\delta,+}(\mathbf{v}) = \big( \{\!\!\{ (C_i(\phi_1(\mathbf{v}, w)), \#_i^1(\mathbf{v}, \phi_1(\mathbf{v}, w))) \mid w \in N(v_1) \}\!\!\}, \ldots, \\ \{\!\!\{ (C_i(\phi_k(\mathbf{v}, w)), \#_i^k(\mathbf{v}, \phi_k(\mathbf{v}, w))) \mid w \in N(v_k) \}\!\!\} \big), \tag{4}$$

instead of $\delta$-$k$-LWL aggregation defined in Equation (3). Here, the function

$$\#_i^j(\mathbf{v}, \mathbf{x}) = \big| \{ \mathbf{w} \colon \ \mathbf{w} \sim_j \mathbf{v}, \ C_i(\mathbf{w}) = C_i(\mathbf{x}) \} \big|, \tag{5}$$

where $\mathbf{w} \sim_j \mathbf{v}$ denotes that $\mathbf{w}$ is $j$-neighbor of $\mathbf{v}$, for $j$ in $[k]$. Essentially, $\#_i^j(\mathbf{v}, \mathbf{x})$ counts the number of $j$-neighbors (local or global) of $\mathbf{v}$ which have the same color as $\mathbf{x}$ under the coloring $C_i$ (i.e., after $i$ rounds). For a fixed $\mathbf{v}$, the function $\#_i^j(\mathbf{v}, \cdot)$ is uniform over the set $S \cap N_j$, where $S$ is a color class obtained after $i$ iterations of the $\delta$-$k$-LWL$^+$ and $N_j$ denotes the set of $j$-neighbors of $\mathbf{v}$. Note that after the stable partition has been reached $\#_i^j(\mathbf{v})$ will not change anymore. Intuitively, this variant captures local and to some extent global information, while still taking the sparsity of the underlying graph into

account. Moreover, observe that each iteration of the $\delta$-$k$-LWL$^+$ has the same asymptotic running time as an iteration of the $\delta$-$k$-LWL, and that the information of the # function is already implicitly contained in Equation (2).

The following theorem shows that the local variant $\delta$-$k$-LWL$^+$ is at least as powerful as the $\delta$-$k$-WL when restricted to the class of connected graphs. The possibly slower convergence leads to advantages in a machine learning setting, see Sections 4 and 6, and also Section 5 for a discussion of practicality, running times, and remaining challenges.

**Theorem 2.** For the class of connected graphs, the following holds for all $k \geq 1$:

$$\delta\text{-}k\text{-LWL}^+ \equiv \delta\text{-}k\text{-WL}.$$

Along with Proposition 1, this establishes the superiority of the $\delta$-$k$-LWL$^+$ over the $k$-WL.

**Corollary 3.** For the class of connected graphs, the following holds for all $k \geq 2$:

$$\delta\text{-}k\text{-LWL}^+ \sqsubset k\text{-WL}.$$

In fact, the proof of Proposition 1 shows that the infinite family of graphs $G_k, H_k$ witnessing the strictness condition can even be distinguished by the $\delta$-$k$-LWL, for each corresponding $k \geq 2$. We note here that the restriction to connected graphs can easily be circumvented by adding a specially marked vertex, which is connected to every other vertex in the graph.

**Kernels based on vertex refinement algorithms** After running the $\delta$-$k$-LWL (and the other vertex refinements algorithms), the concatenation of the histogram of colors in each iteration can be used as a feature vector in a kernel computation. Specifically, in the histogram for every color $c$ in $\mathbb{N}$ there is an entry containing the number of nodes or $k$-tuples that are colored with $c$.

**Local converges to global: proof of Theorem 2** The main technique behind the proof is to construct tree-representations of the colors assigned by the $k$-WL (or its variants). Given a graph $G$, a tuple **v**, and an integer $\ell \geq 0$, the *unrolling tree* of the graph $G$ at **v** *of depth* $\ell$ is a rooted directed tree UNR $[G, \mathbf{s}, \ell]$ (with vertex and edge labels) which encodes the color assigned by $k$-WL to the tuple **v** after $\ell$ rounds, see Appendix D.2 for a formal definition and Figure 4 for an illustration. The usefulness of these tree representations is established by the following lemma. Formally, let **s** and **t** be two $k$-vertex-tuples in $V(G)^k$.

**Lemma 4.** The colors of **s** and **t** after $\ell$ rounds of $k$-WL are identical if and only if the unrolling tree UNR $[G, \mathbf{s}, \ell]$ is isomorphic to the unrolling tree UNR $[G, \mathbf{t}, \ell]$.

For different $k$-WL variants, the construction of these unrollings are slightly different, since an unrolling tree needs to faithfully represent the corresponding aggregation process for generating new colors. For the variants $\delta$-$k$-WL, $\delta$-$k$-LWL, and $\delta$-$k$-LWL$^+$, we define respective unrolling trees $\delta$-UNR $[G, \mathbf{s}, \ell]$, L-UNR $[G, \mathbf{s}, \ell]$, and L$^+$-UNR $[G, \mathbf{s}, \ell]$ along with analogous lemmas, as above, stating their correctness/usefulness. Finally, we show that for *connected graphs*, the $\delta$-UNR unrolling trees (of sufficiently large depth) at two tuples **s** and **t** are identical only if the respective $\delta$-$k$-LWL$^+$ unrolling trees (of sufficiently larger depth) are identical, as shown in the following lemma.

**Lemma 5.** Let $G$ be a connected graph, and let **s** and **t** in $V(G)^k$. If the stable colorings of **s** and **t** under $\delta$-$k$-LWL$^+$ are identical, then the stable colorings of **s** and **t** under $\delta$-$k$-WL are also identical.

Hence, the local algorithm $\delta$-$k$-LWL$^+$ is at least as powerful as the global $\delta$-$k$-WL, for connected graphs, i.e., $\delta$-$k$-LWL$^+ \sqsubseteq \delta$-$k$-WL. The exact details and parameters of this proof can be found in the Appendix.

## 4 Higher-order neural architectures

Although the discrete kernels defined in the previous section are quite powerful, they are limited due to their fixed feature construction scheme, hence suffering from poor adaption to the learning task at hand and the inability to handle continuous node and edge labels in a meaningful way. Moreover, they often result in high-dimensional embeddings forcing one to resort to non-scalable, kernelized optimization procedures. This motivates our definition of a new neural architecture, called *local $\delta$-$k$-GNN ($\delta$-$k$-LGNN)*. Given a labeled graph $G$, let each tuple **v** in $V(G)^k$ be annotated with an

initial feature $f^{(0)}(\mathbf{v})$ determined by its isomorphism type. In each layer $t > 0$, we compute a new feature $f^{(t)}(\mathbf{v})$ as

$$f_{\text{mrg}}^{W_1}\Big(f^{(t-1)}(\mathbf{v}), f_{\text{agg}}^{W_2}\big(\{\!\{f^{(t-1)}(\phi_1(\mathbf{v}, w)) \mid w \in \delta(v_1)\}\!\}, \ldots, \{\!\{f^{(t-1)}(\phi_k(\mathbf{v}, w)) \mid w \in \delta(v_k)\}\!\}\big)\Big),$$

in $\mathbb{R}^{1 \times e}$ for a tuple $\mathbf{v}$, where $W_1^{(t)}$ and $W_2^{(t)}$ are learnable parameter matrices from $\mathbb{R}^{d \times e}$.[5] Moreover, $f_{\text{mrg}}^{W_2}$ and the permutation-invariant $f_{\text{agg}}^{W_1}$ can be arbitrary (permutation-invariant) differentiable functions, responsible for merging and aggregating the relevant feature information, respectively. Initially, we set $f^{(0)}(\mathbf{v})$ to a one-hot encoding of the (labeled) isomorphism type of $G[\mathbf{v}]$. Note that we can naturally handle discrete node and edge labels as well as directed graphs, see Section 4 on how to deal with continuous information. The following result demonstrates the expressive power of the $\delta$-$k$-GNN, in terms of distinguishing non-isomorphic graphs.

**Theorem 6.** Let $(G, l)$ be a labeled graph. Then for all $t \geq 0$ there exists a sequence of weights $\mathbf{W}^{(t)}$ such that

$$C_t^{k,\delta}(\mathbf{v}) = C_t^{k,\delta}(\mathbf{w}) \iff f^{(t)}(\mathbf{v}) = f^{(t)}(\mathbf{w}).$$

Hence, for all graphs, the following holds for all $k \geq 1$:

$$\delta\text{-}k\text{-LGNN} \equiv \delta\text{-}k\text{-LWL}.$$

Moreover, the $\delta$-$k$-GNN inherits the main strength of the $\delta$-$k$-LWL, i.e., it can be implemented using sparse matrix multiplication. Note that it is not possible to come up with an architecture, i.e., instantiations of $f_{\text{mrg}}^{W_1}$ and $f_{\text{agg}}^{W_2}$, such that it becomes more powerful than the $\delta$-$k$-LWL, see [83]. However, all results from the previous section can be lifted to the neural setting. That is, one can derive neural architectures based on the $\delta$-$k$-LWL$^+$, $\delta$-$k$-WL, and $k$-WL, called $\delta$-$k$-LGNN$^+$, $\delta$-$k$-GNN, and $k$-WL-GNN, respectively, and prove results analogous to Theorem 6.

**Incorporating continous information** Since many real-world graphs, e.g., molecules, have continuous features (real-valued vectors) attached to vertices and edges, using a one-hot encoding of the (labeled) isomorphism type is not a sensible choice. Let $a \colon V(G) \to \mathbb{R}^{1 \times d}$ be a function such that each vertex $v$ is annotated with a feature $a(v)$ in $\mathbb{R}^{1 \times d}$, and let $\mathbf{v} = (v_1, \ldots, v_k)$ be a $k$-tuple of vertices. Then we can compute an inital feature

$$f^{(0)}(\mathbf{v}) = f_{\text{enc}}^{W_3}\big((a(v_1), \ldots, a(v_k))\big), \tag{6}$$

for the tuple $\mathbf{v}$. Here, $f_{\text{enc}} \colon \big(\mathbb{R}^{1 \times d}\big)^k \to \mathbb{R}^{1 \times e}$ is an arbitrary differentiable, parameterized function, e.g., a multi-layer perceptron or a standard GNN aggregation function, that computes a joint representation of the $k$ node features $a(v_1), \ldots, a(v_k)$. Moreover, it is also straightforward to incorporate the labeled isomorphism type and continuous edge label information. We further explore this in the experimental section.

**Generalization abilities of the neural architecture** Garg et al. [41], studied the generalization abilities of a standard GNN architecture for binary classification using a margin loss. Under mild conditions, they bounded the empirical Rademacher complexity as $\tilde{\mathcal{O}}\big(rdL/\sqrt{m}\gamma\big)$, where $d$ is the maximum degree of the employed graphs, $r$ is the number of components of the node features, $L$ is the number of layers, and $\gamma$ is a parameter of the loss function. It is straightforward to transfer the above bound to the higher-order (local) layer from above. Hence, this shows that local, sparsity-aware, higher-order variants, e.g., $\delta$-$k$-LGNN, exhibit a smaller generalization error compared to dense, global variants like the $k$-WL-GNN.

## 5 Practicality, barriers ahead, and possible road maps

As Theorem 2 shows, the $\delta$-$k$-LWL$^+$ and its corresponding neural architecture, the $\delta$-$k$-LGNN$^+$, have the same power in distinguishing non-isomorphic graphs as $\delta$-$k$-WL. Although for dense graphs, the local algorithms will have the same running time, for sparse graphs, the running time for each iteration can be upper-bounded by $|n^k| \cdot kd$, where $d$ denotes the maximum or average degree of the graph. Hence, the local algorithm takes the sparsity of the underlying graph into account, resulting in improved computation times compared to the non-local $\delta$-$k$-WL and the $k$-WL (for the same number

of iterations). These observations also translate into practice, see Section 6. The same arguments can be used in favor of the $\delta$-$k$-LWL and $\delta$-$k$-LGNN, which lead to even sparser algorithms.

**Obstacles** The biggest obstacle in applying the algorithms to truly large graphs is the fact that the algorithm considers all possible $k$-tuples leading to a lower bound on the running time of $\Omega(n^k)$. Lifting the results to the folklore $k$-WL, e.g., [77], only "shaves off one dimension". Moreover, applying higher-order algorithms for large $k$ might lead to overfitting issues, see also Section 6.

**Possible solutions** Recent sampling-based approaches for graph kernels or GNNs, see, e.g., [22, 23, 51, 54, 82] address the dependence on $n^k$, while appropriate pooling methods along the lines of Equation (6) address the overfitting issue. Finally, new directions from the theory community, e.g., [50] paint further directions, which might result in more scalable algorithms.

## 6    Experimental evaluation

Our intention here is to investigate the benefits of the local, sparse algorithms, both kernel and neural architectures, compared to the global, dense algorithms, and standard kernel and GNN baselines. More precisely, we address the following questions:

**Q1** Do the local algorithms, both kernel and neural architectures, lead to improved classification and regression scores on real-world benchmark datasets compared to global, dense algorithms and standard baselines?

**Q2** Does the $\delta$-$k$-LWL$^+$ lead to improved classification accuracies compared to the $\delta$-$k$-LWL? Does it lead to higher computation times?

**Q3** Do the local algorithms prevent overfitting to the training set?

**Q4** How much do the local algorithms speed up the computation time compared to the non-local algorithms or dense neural architectures?

The source code of all methods and evaluation procedures is available at `https://www.github.com/chrsmrrs/sparsewl`.

**Datasets** To evaluate kernels, we use the following, well-known, small-scale datasets: ENZYMES [98, 13], IMDB-BINARY, IMDB-MULTI [119], NCI1, NCI109 [109], PTC_FM [53], PROTEINS [31, 13], and REDDIT-BINARY [119]. To show that our kernels also scale to larger datasets, we additionally used the mid-scale datasets: YEAST, YEASTH, UACC257, UACC257H, OVCAR-8, OVCAR-8H [117]. For the neural architectures, we used the large-scale molecular regression datasets ZINC [34, 57] and ALCHEMY [21]. To further compare to the (hierarchical) $k$-GNN [83] and $k$-IGN [77], and show the benefits of our architecture in presence of continuous features, we used the QM9 [91, 112] regression dataset.[6] All datasets can be obtained from `http://www.graphlearning.io` [84]. See Appendix E.1 for further details.

**Kernels** We implemented the $\delta$-$k$-LWL, $\delta$-$k$-LWL$^+$, $\delta$-$k$-WL, and $k$-WL kernel for $k$ in $\{2, 3\}$. We compare our kernels to the Weisfeiler-Leman subtree kernel (1-WL) [100], the Weisfeiler-Leman Optimal Assignment kernel (WLOA) [68], the graphlet kernel (GR) [99], and the shortest-path kernel [13] (SP). All kernels were (re-)implemented in C++11. For the graphlet kernel, we counted (labeled) connected subgraphs of size three. We followed the evaluation guidelines outlined in [84]. We also provide precomputed Gram matrices for easier reproducability.

**Neural architectures** We used the GIN and GIN-$\varepsilon$ architecture [115] as neural baselines. For data with (continuous) edge features, we used a 2-layer MLP to map them to the same number of components as the node features and combined them using summation (GINE and GINE-$\varepsilon$). For the evaluation of the neural architectures of Section 4, $\delta$-$k$-LGNN, $\delta$-$k$-GNN, and $k$-WL-GNN, we implemented them using PYTORCH GEOMETRIC [36], using a Python-wrapped C++11 preprocessing routine to compute the computational graphs for the higher-order GNNs.[7] We used the GIN-$\varepsilon$ layer to express $f_{\mathrm{mrg}}^{W_1}$ and $f_{\mathrm{aggr}}^{W_2}$ of Section 4.

See Appendix E.2 for a detailed description of all evaluation protocols and hyperparameter selection routines.

**Results and discussion** In the following we answer questions **Q1** to **Q4**.

| | Method | **Dataset** | | | | | | | |
|---|---|---|---|---|---|---|---|---|---|
| | | ENZYMES | IMDB-BINARY | IMDB-MULTI | NCI1 | NCI109 | PTC_FM | PROTEINS | REDDIT-BINARY |
| Baseline | GR | 29.7 $_{\pm0.6}$ | 58.9 $_{\pm1.0}$ | 39.0 $_{\pm0.8}$ | 66.1 $_{\pm0.4}$ | 66.3 $_{\pm0.2}$ | 61.3 $_{\pm1.1}$ | 71.2 $_{\pm0.6}$ | 60.0 $_{\pm0.2}$ |
| | SP | 40.7 $_{\pm0.9}$ | 58.5 $_{\pm0.4}$ | 39.4 $_{\pm0.3}$ | 74.0 $_{\pm0.3}$ | 73.0 $_{\pm0.4}$ | 61.3 $_{\pm1.3}$ | 75.6 $_{\pm0.5}$ | 84.6 $_{\pm0.3}$ |
| | 1-WL | 50.7 $_{\pm1.2}$ | 72.5 $_{\pm0.5}$ | 50.0 $_{\pm0.5}$ | 84.2 $_{\pm0.3}$ | 84.3 $_{\pm0.3}$ | **62.6** $_{\pm2.0}$ | 72.6 $_{\pm1.2}$ | 72.8 $_{\pm0.5}$ |
| | WLOA | 56.8 $_{\pm1.6}$ | 72.7 $_{\pm0.9}$ | 50.1 $_{\pm0.7}$ | 84.9 $_{\pm0.3}$ | 85.2 $_{\pm0.3}$ | 61.8 $_{\pm1.5}$ | 73.2 $_{\pm0.6}$ | 88.1 $_{\pm0.4}$ |
| Neural | Gin-$0$ | 38.8 $_{\pm1.7}$ | 72.7 $_{\pm0.9}$ | 49.9 $_{\pm0.8}$ | 78.5 $_{\pm0.5}$ | 76.7 $_{\pm0.8}$ | 58.2 $_{\pm3.3}$ | 71.3 $_{\pm0.9}$ | 89.8 $_{\pm0.6}$ |
| | Gin-$\varepsilon$ | 39.4 $_{\pm1.7}$ | 72.9 $_{\pm0.6}$ | 49.6 $_{\pm0.9}$ | 78.6 $_{\pm0.3}$ | 77.0 $_{\pm0.5}$ | 57.7 $_{\pm2.0}$ | 71.1 $_{\pm0.8}$ | 90.3 $_{\pm0.3}$ |
| Global | 2-WL | 36.7 $_{\pm1.7}$ | 68.2 $_{\pm1.1}$ | 48.1 $_{\pm0.5}$ | 67.1 $_{\pm0.3}$ | 67.5 $_{\pm0.2}$ | 62.3 $_{\pm1.6}$ | 75.0 $_{\pm0.8}$ | OOM |
| | 3-WL | 42.3 $_{\pm1.1}$ | 67.8 $_{\pm0.8}$ | 47.0 $_{\pm0.7}$ | OOT | OOT | 61.5 $_{\pm1.7}$ | OOM | OOM |
| | $\delta$-2-WL | 37.5 $_{\pm1.2}$ | 68.1 $_{\pm1.1}$ | 47.9 $_{\pm0.7}$ | 67.0 $_{\pm0.5}$ | 67.2 $_{\pm0.4}$ | 61.9 $_{\pm0.9}$ | 75.0 $_{\pm0.4}$ | OOM |
| | $\delta$-3-WL | 43.0 $_{\pm1.4}$ | 67.5 $_{\pm1.0}$ | 47.3 $_{\pm0/9}$ | OOT | OOT | 61.2 $_{\pm2.0}$ | OOM | OOM |
| Local | $\delta$-2-LWL | 56.6 $_{\pm1.2}$ | 73.3 $_{\pm0.5}$ | 50.2 $_{\pm0.6}$ | 84.7 $_{\pm0.3}$ | 84.2 $_{\pm0.4}$ | 60.3 $_{\pm3.2}$ | 75.1 $_{\pm0.3}$ | 89.7 $_{\pm0.4}$ |
| | $\delta$-2-LWL$^+$ | 52.9 $_{\pm1.4}$ | 75.7 $_{\pm0.7}$ | 62.5 $_{\pm1.0}$ | **91.4** $_{\pm0.2}$ | **89.3** $_{\pm0.2}$ | **62.6** $_{\pm1.6}$ | **79.3** $_{\pm1.1}$ | **91.1** $_{\pm0.5}$ |
| | $\delta$-3-LWL | **57.6** $_{\pm1.2}$ | 72.8 $_{\pm1.2}$ | 49.3 $_{\pm1.0}$ | 83.4 $_{\pm0.2}$ | 82.4 $_{\pm0.4}$ | 61.3 $_{\pm1.6}$ | OOM | OOM |
| | $\delta$-3-LWL$^+$ | 56.8 $_{\pm1.2}$ | **76.2** $_{\pm0.8}$ | **64.2** $_{\pm0.9}$ | 82.7 $_{\pm0.5}$ | 81.9 $_{\pm0.4}$ | 61.3 $_{\pm2.0}$ | OOM | OOM |

Table 1: Classification accuracies in percent and standard deviations, OOT— Computation did not finish within one day, OOM— Out of memory.

(a) Training versus test accuracy of local and global kernels for a subset of the datasets.

| | Set | **Dataset** | | |
|---|---|---|---|---|
| | | ENZYMES | IMDB-BINARY | IMDB-MULTI |
| $\delta$-2-WL | Train | 91.2 | 83.8 | 57.6 |
| | Test | 37.5 | 68.1 | 47.9 |
| $\delta$-2-LWL | Train | 98.8 | 83.5 | 59.9 |
| | Test | 56.6 | 73.3 | 50.2 |
| $\delta$-2-LWL$^+$ | Train | 99.5 | 95.1 | 86.5 |
| | Test | 52.9 | 75.7 | 62.5 |

(b) Mean MAE (mean std. MAE, logMAE) on large-scale (multi-target) molecular regression tasks.

| | Method | **Dataset** | |
|---|---|---|---|
| | | ZINC (FULL) | ALCHEMY (FULL) |
| Baseline | GINE-$\varepsilon$ | 0.084 $_{\pm0.004}$ | 0.103 $_{\pm0.001}$ -2.956 $_{\pm0.029}$ |
| | 2-WL-GNN | 0.133 $_{\pm0.013}$ | 0.093 $_{\pm0.001}$ -3.394 $_{\pm0.035}$ |
| | $\delta$-2-GNN | **0.042** $_{\pm0.003}$ | **0.080** $_{\pm0.001}$ -3.516 $_{\pm0.021}$ |
| | $\delta$-2-LGNN | 0.045 $_{\pm0.006}$ | 0.083 $_{\pm0.001}$ -3.476 $_{\pm0.025}$ |

Table 2: Additional results for kernel and neural approaches.

**A1** *Kernels* See Table 1. The local algorithm, for $k = 2$ and 3, severely improves the classification accuracy compared to the $k$-WL and the $\delta$-$k$-WL. For example, on the ENZYMES dataset the $\delta$-2-LWL achieves an improvement of almost 20%, and the $\delta$-3-LWL achieves the best accuracies over all employed kernels, improving over the 3-WL and the $\delta$-3-WL by more than 13%. This observation holds over all datasets. Our algorithms also perform better than neural baselines. See Table 5 in the appendix for additional results on the mid-scale datasets. However, it has to be noted that increasing $k$ does not always result in increased accuracies. For example, on all datasets (excluding ENZYMES), the performance of the $\delta$-2-LWL is better or on par with the $\delta$-3-LWL. Hence, with increasing $k$ the local algorithm is more prone to overfitting.

*Neural architectures* See Table 2b and Figure 2. On the ZINC and ALCHEMY datasets, the $\delta$-2-LGNN is on par or slightly worse than the $\delta$-2-GNN. Hence, this is in contrast to the kernel variant. We assume that this is due to the $\delta$-2-GNN being more flexible than its kernel variant in weighing the importance of global and local neighbors. This is further highlighted by the worse performance of the 2-WL-GNN, which even performs worse than the (1-dimensional) GINE-$\varepsilon$ on the ZINC dataset. On the QM9 dataset, see Figure 2a, the $\delta$-2-LGNN performs better than the higher-order methods from [77, 83] while being on par with the MPNN architecture. We note here that the MPNN was specifically tuned to the QM9 dataset, which is not the case for the $\delta$-2-LGNN (and the other higher-order architectures).

**A2** See Table 1. The $\delta$-2-LWL$^+$ improves over the $\delta$-2-LWL on all datasets excluding ENZYMES. For example, on IMDB-MULTI, NCI1, NCI109, and PROTEINS the algorithm achieves an improvement over of 4%, respectively, achieving a new state-of-the-art. The computation times are only increased slightly, see Table 8 in the appendix. Similar results can be observed on the mid-scale datasets, see Tables 5 and 9 in the appendix.

**A3** *Kernels* As Table 2a (Table 6 for all datasets) shows the $\delta$-2-WL reaches slightly higher training accuracies over all datasets compared to the $\delta$-2-LWL, while the testing accuracies are much lower (excluding PTC_FM and PROTEINS). This indicates that the $\delta$-2-WL overfits on the training set. The higher test accuracies of the local algorithm are likely due to the smaller neighborhood, which promotes that the number of colors grow slower compared to the global algorithm. The $\delta$-$k$-LWL$^+$

| | Method | QM9 |
|---|---|---|
| Baseline | GINE-$\varepsilon$ | $0.081 \pm 0.003$ |
| | MPNN | $0.034 \pm 0.001$ |
| | 1-2-GNN | $0.068 \pm 0.001$ |
| | 1-3-GNN | $0.088 \pm 0.007$ |
| | 1-2-3-GNN | $0.062 \pm 0.001$ |
| | 3-IGN | $0.046 \pm 0.001$ |
| | $\delta$-2-LGNN | $\mathbf{0.029} \pm 0.001$ |

(a) Mean std. MAE compared to [45, 77, 83].

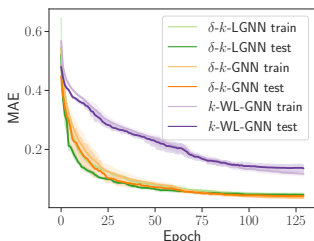

(b) ZINC

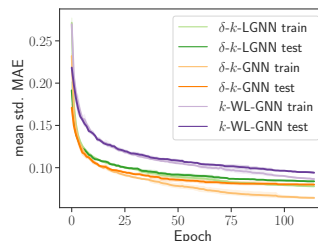

(c) ALCHEMY

Figure 2: Additional results for neural architectures.

(a) Speed up ratios of local kernel computations for a subset of the datasets.

| | Method | Dataset | | |
|---|---|---|---|---|
| | | ENZYMES | IMDB-BINARY | NCI1 |
| Global | 2-WL | 10.4 | 3.6 | 14.3 |
| | $\delta$-2-WL | 10.1 | 3.6 | 14.5 |
| | $\delta$-2-LWL$^+$ | 1.2 | 1.2 | 1.3 |
| | $\delta$-2-LWL | 1.0 | 1.0 | 1.0 |

(b) Average speed up ratios over all epochs (training and testing).

| | Method | Dataset | |
|---|---|---|---|
| | | ZINC | ALCHEMY |
| Dense | 2-WL-GNN | 2.2 | 1.1 |
| | $\delta$-2-GNN | 2.5 | 1.7 |
| | GINE-$\varepsilon$ | 0.2 | 0.4 |
| | $\delta$-2-LGNN | 1.0 | 1.0 |

Table 3: Speed up ratios of local over global algorithms.

inherits the strengths of both algorithms, i.e., achieving the overall best training accuracies while achieving state-of-the-art testing accuracies.

*Neural architectures* See Figure 2. In contrast to the kernel variants, the 2-WL and the $\delta$-2-WL, the corresponding neural architectures, the 2-WL-GNN and the $\delta$-2-GNN, seem less prone to overfitting. However, especially on the ALCHEMY dataset, the $\delta$-2-LGNN overfits less.

**A4** *Kernels* See Table 3a (Tables 8 and 9 for all datasets). The local algorithm severely speeds up the computation time compared to the $\delta$-$k$-WL and the $k$-WL for $k = 2$ and 3. For example, on the ENZYMES dataset the $\delta$-2-LWL is over ten times faster than the $\delta$-2-WL. The improvement of the computation times can be observed across all datasets. For some datasets, the $\{2, 3\}$-WL and the $\delta$-$\{2, 3\}$-WL did not finish within the given time limit or went out of memory. For example, on four out of eight datasets, the $\delta$-3-WL is out of time or out of memory. In contrast, for the corresponding local algorithm, this happens only two out of eight times. Hence, the local algorithm is more suitable for practical applications.

*Neural architectures* See Table 3b. The local algorithm severely speeds up the computation time of training and testing. Especially, on the ZINC dataset, which has larger graphs compared to the ALCHEMY dataset, the $\delta$-2-LGNN achieves a computation time that is more than two times lower compared to the $\delta$-2-GNN and the 2-WL-GNN.

# 7 Conclusion

We introduced local variants of the $k$-dimensional Weisfeiler-Leman algorithm. We showed that one variant and its corresponding neural architecture are strictly more powerful than the $k$-WL while taking the underlying graph's sparsity into account. To demonstrate the practical utility of our findings, we applied them to graph classification and regression. We verified that our local, sparse algorithms lead to vastly reduced computation times compared to their global, dense counterparts while establishing new state-of-the-art results on a wide range of benchmark datasets. *We believe that our local, higher-order kernels and GNN architectures should become a standard approach in the regime of supervised learning with small graphs, e.g., molecular learning.*

Future work includes a more fine-grained analysis of the proposed algorithm, e.g., moving away from the restrictive graph isomorphism objective and deriving a deeper understanding of the neural architecture's capabilities when optimized with stochastic gradient descent.

## Broader impact

We view our work mainly as a methodological contribution. It studies the limits of current (supervised) graph embeddings methods, commonly used in chemoinformatics [103], bioinformatics [10], or network science [27]. Currently, methods used in practice, such as GNNs or extended-connectivity fingerprints [93] have severe limitations and might miss crucial patterns in today's complex, interconnected data. We investigate how to scale up graph embeddings that can deal with higher-order interactions of vertices (or atom of molecules, users in social networks, variables in optimization, ...) to larger graphs or networks. Hence, our method paves the way for more resource-efficient and expressive graph embeddings.

We envision that our (methodological) contributions enable the design of more expressive and scalable graph embeddings in fields such as quantum chemistry, drug-drug interaction prediction, in-silicio, data-driven drug design/generation, and network analysis for social good. However, progress in graph embeddings might also trigger further advancements in hostile social network analysis, e.g., extracting more fine-grained user interactions for social tracking.

**Example impact** We are actively cooperating with chemists on drug design to evaluate further our approach to new databases for small molecules. Here, the development of new databases is quite tedious, and graph embeddings can provide hints to the wet lab researcher where to start their search. However, still, humans need to do much of the intuition-driven, manual wet lab work. Hence, we do not believe that our methods will result in job losses in the life sciences in the foreseeable future.

## Acknowledgments and disclosure of funding

We thank Matthias Fey for answering our questions with regard to PYTORCH GEOMETRIC and Xavier Bresson for providing the ZINC dataset. This work is funded by the Deutsche Forschungsgemeinschaft (DFG, German Research Foundation) under Germany's Excellence Strategy – EXC-2047/1 – 390685813 and under DFG Research Grants Program–RA 3242/1-1–411032549.

## Footnotes

[4]We define the 1-WL in the next subsection.

[5]For clarity of presentation we omit biases.

[6]We opted for comparing on the QM9 dataset to ensure a fair comparison concerning hyperparameter selection.

[7]We opted for not implementing the $\delta$-$k$-LGNN$^+$ as it would involve precomputing #.

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
