[Supplementary Material]

# Appendix

## A  Related work (Expanded)

In the following, we review related work from graph kernels, GNNs, and theory.

**Graph kernels** Historically, kernel methods—which implicitly or explicitly map graphs to elements of a Hilbert space—have been the dominant approach for supervised learning on graphs. Important early work in this area includes random-walk based kernels [42, 60, 70] and kernels based on shortest paths [13]. More recently, graph kernels' developments have emphasized scalability, focusing on techniques that bypass expensive Gram matrix computations by using explicit feature maps, see, e.g., [100]. Morris et al. [82] devised a local, set-based variant of the $k$-WL. However, the approach is (provably) weaker than the tuple-based algorithm, and they do not prove convergence to the original algorithm. Yanardag and Vishwanathan successfully employed Graphlet [99], and Weisfeiler-Leman kernels within frameworks for smoothed [118] and deep graph kernels [119]. Other recent works focus on assignment-based [59, 68, 88], spectral [67, 106], graph decomposition [89], randomized binning approaches [52], and the extension of kernels based on the 1-WL [92, 104]. For a theoretical investigation of graph kernels, see [69], for a thorough survey of graph kernels, see [71].

**GNNs** Recently, graph neural networks (GNNs) [45, 97] emerged as an alternative to graph kernels. Notable instances of this architecture include, e.g., [33, 37, 51, 105], and the spectral approaches proposed in, e.g., [14, 29, 64, 81]—all of which descend from early work in [65, 80, 102, 97]. Recent extensions and improvements to the GNN framework include approaches to incorporate different local structures (around subgraphs), e.g., [1, 38, 58, 87, 114], novel techniques for pooling node representations in order perform graph classification, e.g., [16, 40, 120, 122], incorporating distance information [121], and non-euclidian geometry approaches [18]. Moreover, recently empirical studies on neighborhood aggregation functions for continuous vertex features [26], edge-based GNNs leveraging physical knowledge [2, 66], and sparsification methods [94] emerged. Loukas [74] and Sato et al. studied the limits of GNNs when applied to combinatorial problems. A survey of recent advancements in GNN techniques can be found, e.g., in [19, 113, 125]. Garg et al. [41] and Verma and Zhang [107] studied the generalization abilities of GNNs, and [32] related wide GNNs to a variant of the neural tangent kernel [3, 56]. Murphy et al. [85, 86] and Sato et al. [96] extended the expressivity of GNNs by considering all possible permutations of a graph's adjacency matrix, or adding random node features, respectivley. The connection between random colorings and universality was investigated in [28].

Recently, connections to Weisfeiler-Leman type algorithms have been shown [11, 24, 43, 44, 75, 77, 83, 115]. Specifically, [83, 115] showed that the expressive power of any possible GNN architecture is limited by the 1-WL in terms of distinguishing non-isomorphic graphs. Morris et al. [83] also introduced $k$-*dimensional GNNs* ($k$-GNN) which rely on a message-passing scheme between subgraphs of cardinality $k$. Similar to [82], the paper employed a local, set-based (neural) variant of the $k$-WL, which is (provably) weaker than the variant considered here. Later, this was refined in [77] by introducing $k$-*order invariant graph networks* ($k$-IGN), based on Maron et al. [78], and references therein, which are equivalent to the folklore variant of the $k$-WL [46] in terms of distinguishing non-isomorphic graphs. However, $k$-IGN may not scale since they rely on dense linear algebra routines. Chen et al. [24] connect the theory of universal approximation of permutation-invariant functions and the graph isomorphism viewpoint and introduce a variation of the 2-WL, which is more powerful than the former. Our comprehensive treatment of higher-order, sparse, neural networks for arbitrary $k$ subsumes all of the algorithms and neural architectures mentioned above.

Finally, there exists a new line of work focusing on extending GNNs to hypergraphs, see, e.g., [9, 116, 123], and a line of work in the data mining community incorporating global or higher-order information into graph or node embeddings, see, e.g., [17, 72, 79].

**Theory** The Weisfeiler-Leman algorithm constitutes one of the earliest approaches to isomorphism testing [110, 111], having been heavily investigated by the theory community over the last few decades [49]. Moreover, the fundamental nature of the $k$-WL is evident from a variety of connections to other fields such as logic, optimization, counting complexity, and quantum computing. The power and limitations of $k$-WL can be neatly characterized in terms of logic and descriptive complexity [55], Sherali-Adams relaxations of the natural integer linear program for the graph isomorphism problem [6, 48, 76], homomorphism counts [30], and quantum isomorphism games [7]. In their seminal

paper [55], Cai et al. showed that for each $k$ there exists a pair of non-isomorphic graphs of size $\mathcal{O}(k)$ each that cannot be distinguished by the $k$-WL. Grohe et al. [49] gives a thorough overview of these results. For $k = 1$, the power of the algorithm has been completely characterized [4, 63]. Moreover, upper bounds on the running time for $k = 1$ [12, 61], and the number of iterations for the folklore $k = 2$ [62, 73] have been shown. For $k = 1$ and 2, Arvind et al. [5] studied the abilities of the (folklore) $k$-WL to detect and count fixed subgraphs, extending the work of Fürer [39]. The former was refined in [25]. The algorithm (for logarithmic $k$) plays a prominent role in the recent result of Babai [8] improving the best-known running time for the graph isomorphism problem. Recently, Grohe et al. [50] introduced the framework of Deep Weisfeiler Leman algorithms, which allow the design of a more powerful graph isomorphism test than Weisfeiler-Leman type algorithms. Finally, the emerging connections between the Weisfeiler-Leman paradigm and graph learning are described in a recent survey of Grohe [47].

## B    Preliminaries (Expanded)

We briefly describe the Weisfeiler-Leman algorithm and, along the way, introduce our notation. We also state a variant of the algorithm, introduced in [76]. As usual, let $[n] = \{1, \ldots, n\} \subset \mathbb{N}$ for $n \geq 1$, and let $\{\!\{\ldots\}\!\}$ denote a multiset.

**Graphs** A *graph* $G$ is a pair $(V, E)$ with a *finite* set of *vertices* $V$ and a set of *edges* $E \subseteq \{\{u, v\} \subseteq V \mid u \neq v\}$. We denote the set of vertices and the set of edges of $G$ by $V(G)$ and $E(G)$, respectively. For ease of notation, we denote the edge $\{u, v\}$ in $E(G)$ by $(u, v)$ or $(v, u)$. In the case of *directed graphs* $E \subseteq \{(u, v) \in V \times V \mid u \neq v\}$. A *labeled graph* $G$ is a triple $(V, E, l)$ with a label function $l \colon V(G) \cup E(G) \to \Sigma$, where $\Sigma$ is some finite alphabet. Then $l(v)$ is a *label* of $v$ for $v$ in $V(G) \cup E(G)$. The *neighborhood* of $v$ in $V(G)$ is denoted by $\delta(v) = N(v) = \{u \in V(G) \mid (v, u) \in E(G)\}$. Moreover, its complement $\bar{\delta}(v) = \{u \in V(G) \mid (v, u) \notin E(G)\}$. Let $S \subseteq V(G)$ then $G[S] = (S, E_S)$ is the *subgraph induced* by $S$ with $E_S = \{(u, v) \in E(G) \mid u, v \in S\}$. A *tree* is a connected graph without cycles. A *rooted tree* is a tree with a designated vertex called *root* in which the edges are directed in such a way that they point away from the root. Let $p$ be a vertex in a directed tree then we call its out-neighbors *children* with parent $p$.

We say that two graphs $G$ and $H$ are *isomorphic* if there exists an edge preserving bijection $\varphi \colon V(G) \to V(H)$, i.e., $(u, v)$ is in $E(G)$ if and only if $(\varphi(u), \varphi(v))$ is in $E(H)$. If $G$ and $H$ are isomorphic, we write $G \simeq H$ and call $\varphi$ an *isomorphism* between $G$ and $H$. Moreover, we call the equivalence classes induced by $\simeq$ *isomorphism types*, and denote the isomorphism type of $G$ by $\tau_G$. In the case of labeled graphs, we additionally require that $l(v) = l(\varphi(v))$ for $v$ in $V(G)$ and $l((u, v)) = l((\varphi(u), \varphi(v)))$ for $(u, v)$ in $E(G)$. Let $\mathbf{v}$ be a *tuple* in $V(G)^k$ for $k > 0$, then $G[\mathbf{v}]$ is the subgraph induced by the components of $\mathbf{v}$, where the vertices are labeled with integers from $\{1, \ldots, k\}$ corresponding to indices of $\mathbf{v}$.

**Kernels** A *kernel* on a non-empty set $\mathcal{X}$ is a positive semidefinite function $k \colon \mathcal{X} \times \mathcal{X} \to \mathbb{R}$. Equivalently, a function $k$ is a kernel if there is a *feature map* $\phi \colon \mathcal{X} \to \mathcal{H}$ to a Hilbert space $\mathcal{H}$ with inner product $\langle \cdot, \cdot \rangle$, such that $k(x, y) = \langle \phi(x), \phi(y) \rangle$ for all $x$ and $y$ in $\mathcal{X}$. Let $\mathcal{G}$ be the set of all graphs, then a (positive semidefinite) function $\mathcal{G} \times \mathcal{G} \to \mathbb{R}$ is called a *graph kernel*.

## C    Vertex refinement algorithms (Expanded)

Let $k$ be a fixed positive integer. As usual, let $V(G)^k$ denote the set of $k$-tuples of vertices of $G$.

A *coloring* of $V(G)^k$ is a mapping $C \colon V(G)^k \to \mathbb{N}$, i.e., we assign a number (color) to every tuple in $V(G)^k$. The *initial coloring* $C_0$ of $V(G)^k$ is specified by the isomorphism types of the tuples, i.e., two tuples $\mathbf{v}$ and $\mathbf{w}$ in $V(G)^k$ get a common color iff the mapping $v_i \to w_i$ induces an isomorphism between the labeled subgraphs $G[\mathbf{v}]$ and $G[\mathbf{w}]$. A *color class* corresponding to a color $c$ is the set of all tuples colored $c$, i.e., the set $C^{-1}(c)$.

The *neighborhood* of a vertex tuple $\mathbf{v}$ in $V(G)^k$ is defined as follows. For $j$ in $[k]$, let $\phi_j(\mathbf{v}, w)$ be the $k$-tuple obtained by replacing the $j^{\text{th}}$ component of $\mathbf{v}$ with the vertex $w$. That is, $\phi_j(\mathbf{v}, w) = (v_1, \ldots, v_{j-1}, w, v_{j+1}, \ldots, v_k)$. If $\mathbf{w} = \phi_j(\mathbf{v}, w)$ for some $w$ in $V(G)$, call $\mathbf{w}$ a *$j$-neighbor* of $\mathbf{v}$. The neighborhood of $\mathbf{v}$ is thus defined as the set of all tuples $\mathbf{w}$ such that $\mathbf{w} = \phi_j(\mathbf{v}, w)$ for some $j$ in $[k]$ and $w$ in $V(G)$.

The *refinement* of a coloring $C\colon V(G)^k \to \mathbb{N}$, denoted by $\widehat{C}$, is a coloring $\widehat{C}\colon V(G)^k \to \mathbb{N}$ defined as follows. For each $j$ in $[k]$, collect the colors of the $j$-neighbors of $\mathbf{v}$ as a multiset $S_j = \{\!\{C(\phi_j(\mathbf{v}, w)) \mid w \in V(G)\}\!\}$. Then, for a tuple $\mathbf{v}$, define

$$\widehat{C}(\mathbf{v}) = (C(\mathbf{v}), M(\mathbf{v})),$$

where $M(\mathbf{v})$ is the $k$-tuple $(S_1, \ldots, S_k)$. For consistency, the strings $\widehat{C}(\mathbf{v})$ thus obtained are lexicographically sorted and renamed as integers. Observe that the new color $\widehat{C}(\mathbf{v})$ of $\mathbf{v}$ is solely dictated by the color histogram of its neighborhood. In general, a different mapping $M(\cdot)$ could be used, depending on the neighborhood information that we would like to aggregate. We will refer to a mapping $M(\cdot)$ as an *aggregation map*.

**$k$-dimensional Weisfeiler-Leman** For $k \geq 2$, the $k$-WL computes a coloring $C_\infty\colon V(G)^k \to \mathbb{N}$ of a given graph $G$, as follows.[1] To begin with, the initial coloring $C_0$ is computed. Then, starting with $C_0$, successive refinements $C_{i+1} = \widehat{C_i}$ are computed until convergence. That is,

$$C_{i+1}(\mathbf{v}) = (C_i(\mathbf{v}), M_i(\mathbf{v})),$$

where

$$M_i(\mathbf{v}) = \big(\{\!\{C_i(\phi_1(\mathbf{v}, w)) \mid w \in V(G)\}\!\}, \ldots, \{\!\{C_i(\phi_k(\mathbf{v}, w)) \mid w \in V(G)\}\!\}\big). \tag{1}$$

The successive refinement steps are also called *rounds* or *iterations*. Since the disjoint union of the color classes form a partition of $V(G)^k$, there must exist a finite $\ell \leq |V(G)|^k$ such that $C_\ell = \widehat{C_\ell}$. In the end, the $k$-WL outputs $C_\ell$ as the *stable coloring* $C_\infty$.

The $k$-WL *distinguishes* two graphs $G$ and $H$ if, upon running the $k$-WL on their disjoint union $G \dot{\cup} H$, there exists a color $c$ in $\mathbb{N}$ in the stable coloring such that the corresponding color class $S_c$ satisfies

$$|V(G)^k \cap S_c| \neq |V(H)^k \cap S_c|,$$

i.e., there exist an unequal number of $c$-colored tuples in $V(G)^k$ and $V(H)^k$. Hence, two graphs distinguished by the $k$-WL must be non-isomorphic.

In fact, there exist several variants of the above defined $k$-WL. These variants result from the application of different aggregation maps $M(\cdot)$. For example, setting $M(\cdot)$ to be

$$M^F(\mathbf{v}) = \{\!\{\big(C(\phi_1(\mathbf{v}, w)), \ldots, C(\phi_k(\mathbf{v}, w))\big) \mid w \in V(G)\}\!\},$$

yields a well-studied variant of the $k$-WL (see, e.g., [15]), commonly known as "folklore" $k$-WL in machine learning literature. It holds that the $k$-WL using Equation (7) is as powerful as the folklore $(k-1)$-WL [48].

### C.1 $\delta$-Weisfeiler-Leman algorithm

Let $\mathbf{w} = \phi_j(\mathbf{v}, w)$ be a $j$-neighbor of $\mathbf{v}$. Call $\mathbf{w}$ a *local* $j$-neighbor of $\mathbf{v}$ if $w$ is adjacent to the replaced vertex $v_j$. Otherwise, call $\mathbf{w}$ a *global* $j$-neighbor of $\mathbf{v}$. Figure 3 illustrates this definition for a 3-tuple $(u, v, w)$. For tuples $\mathbf{v}$ and $\mathbf{w}$ in $V(G)^k$, the function

$$\mathrm{adj}((\mathbf{v}, \mathbf{w})) = \begin{cases} \mathrm{L} & \text{if } \mathbf{w} \text{ is a local neighbor of } \mathbf{v} \\ \mathrm{G} & \text{if } \mathbf{w} \text{ is a global neighbor of } \mathbf{v} \end{cases}$$

indicates whether $\mathbf{w}$ is a local or global neighbor of $\mathbf{v}$.

The *$\delta$-$k$-dimensional Weisfeiler-Leman algorithm*, denoted by $\delta$-$k$-WL, is a variant of the classic $k$-WL which *differentiates* between the local and the global neighbors during neighborhood aggregation [76]. Formally, the $\delta$-$k$-WL algorithm refines a coloring $C_i$ (obtained after $i$ rounds) via the aggregation map

$$\begin{aligned} M_i^{\delta, \overline{\delta}}(\mathbf{v}) = \big(&\{\!\{(C_i(\phi_1(\mathbf{v}, w)), \mathrm{adj}(\mathbf{v}, \phi_1(\mathbf{v}, w))) \mid w \in V(G)\}\!\}, \ldots, \\ &\{\!\{(C_i(\phi_k(\mathbf{v}, w)), \mathrm{adj}(\mathbf{v}, \phi_k(\mathbf{v}, w))) \mid w \in V(G)\}\!\}\big), \end{aligned} \tag{2}$$

instead of the $k$-WL aggregation specified by Equation (7). We define the 1-WL to be the $\delta$-1-WL, which is commonly known as color refinement or naive vertex classification.

(a) Underlying graph $G$, with tuple $(u, v, w)$

(b) $(u, v, x)$ is a local 3-neighbor of $(u, v, w)$

(c) $(x, v, w)$ is a global 1-neighbor of $(u, v, w)$

Figure 1: Illustration of the local and global neighborhood of the 3-tuple $(u, v, w)$.

**Comparing $k$-WL variants** Given that there exist several variants of the $k$-WL, corresponding to different aggregation maps $M(\cdot)$, it is natural to ask whether they are equivalent in power, vis-a-vis distinguishing non-isomorphic graphs. Let $A_1$ and $A_2$ denote two vertex refinement algorithms, we write $A_1 \sqsubseteq A_2$ if $A_1$ distinguishes between all non-isomorphic pairs $A_2$ does, and $A_1 \equiv A_2$ if both directions hold. The corresponding strict relation is denoted by $\sqsubset$.

The following result relates the power of the $k$-WL and $\delta$-$k$-WL. Since for a graph $G = (V, E)$, $M_i^{\delta, \overline{\delta}}(\mathbf{v}) = M_i^{\delta, \overline{\delta}}(\mathbf{w})$ implies $M_i(\mathbf{v}) = M_i(\mathbf{w})$ for all $\mathbf{v}$ and $\mathbf{w}$ in $V(G)^k$ and $i \geq 0$, it immediately follows that $\delta$-$k$-WL $\sqsubseteq$ $k$-WL. For $k = 1$, these two algorithms are equivalent by definition. For $k \geq 2$, this relation can be shown to be strict, see the next section.

**Proposition 1** (restated, Proposition 1 in the main text). *For all graphs and $k \geq 2$, the following holds:*

$$\delta\text{-}k\text{-WL} \sqsubset k\text{-WL}.$$

### C.1.1  Proof of Proposition 1

It suffices to show an infinite family of graphs $(G_k, H_k)$, $k \in \mathbb{N}$, such that (a) $k$-WL does not distinguish $G_k$ and $H_k$, although (b) $\delta$-$k$-WL distinguishes $G_k$ and $H_k$.

We proceed to the construction of this family. The graph family is based on the classic construction of [15], commonly referred to as Cai-Furer-Immermman (CFI) graphs.

**Construction.** Let $K$ denote the complete graph on $k + 1$ vertices (there are no loops in $K$). The vertices of $K$ are numbered from 0 to $k$. Let $E(v)$ denote the set of edges incident to $v$ in $K$: clearly, $|E(v)| = k$ for all $v \in V(K)$. Define the graph $G$ as follows:

1. For the vertex set $V(G)$, we add
    (a) $(v, S)$ for each $v \in V(K)$ and for each *even* subset $S$ of $E(v)$,
    (b) two vertices $e^1, e^0$ for each edge $e \in E(K)$.
2. For the edge set $E(G)$, we add
    (a) an edge $\{e^0, e^1\}$ for each $e \in E(K)$,
    (b) an edge between $(v, S)$ and $e^1$ if $v \in e$ and $e \in S$,
    (c) an edge between $(v, S)$ and $e^0$ if $v \in e$ and $e \notin S$,

Define a companion graph $H$, in a similar manner to $G$, with the following exception: in Step 1(a), for the vertex $0 \in V(K)$, we choose all *odd* subsets of $E(0)$. Counting vertices, we find that $|V(G)| = |V(H)| = (k + 1) \cdot 2^{k-1} + \binom{k}{2} \cdot 2$. This finishes the construction of graphs $G$ and $H$. We set $G_k := G$ and $H_k := H$.

A set $S$ of vertices is said to form a *distance-two-clique* if the distance between any two vertices in $S$ is exactly two.

**Lemma 2.** *The following holds for graphs $G$ and $H$ defined above.*

- *There exists a distance-two-clique of size $(k + 1)$ inside $G$.*
- *There does not exist a distance-two-clique of size $(k + 1)$ inside $H$.*

Hence, $G$ and $H$ are non-isomorphic.

*Proof.* In the graph $G$, consider the vertex subset $S = \{(0, \emptyset), (1, \emptyset), \ldots, (k, \emptyset)\}$ of size $(k+1)$. That is, from each "cloud" of vertices of the form $(v, S)$ for a fixed $v$, we pick the vertex corresponding to the trivial even subset, the empty set denoted by $\emptyset$. Observe that any two vertices in $S$ are at distance two from each other. This holds because for any $i, j \in V(K)$, $(i, \emptyset)$ is adjacent to $\{i, j\}^0$ which is adjacent to $(j, \emptyset)$ (e.g. see Figure 1). Hence, the vertices in $S$ form a distance-two-clique of size $k+1$.

On the other hand, for the graph $H$, suppose there exists a distance-two-clique, say $(0, S_0), \ldots, (k, S_k)$ in $H$, where each $S_i \subseteq E(i)$. If we compute the parity-sum of the parities of $|S_0|, \ldots, |S_k|$, we end up with 1 since there is exactly one odd subset in this collection, viz. $S_0$. On the other hand, we can also compute this parity-sum in an edge-by-edge manner: for each edge $(i, j) \in E(K)$, since $(i, S_i)$ and $(j, S_j)$ are at distance two, either both $S_i$ and $S_j$ contain the edge $\{i, j\}$ or neither of them contains $\{i, j\}$: hence, the parity-sum contribution of $S_i$ and $S_j$ to the term corresponding to $e$ is zero. Since the contribution of each edge to the total parity-sum is 0, the total parity-sum must be zero. This is a contradiction, and hence, there does not exist a distance-two-clique in $H$. $\qquad\square$

Next, we show that the local algorithm $\delta$-$k$-LWL can distinguish $G$ and $H$. Since $\delta$-$k$-WL $\sqsubseteq \delta$-$k$-LWL, the above lemma implies the strictness condition $\delta$-$k$-WL $\sqsubset k$-WL.

**Lemma 3.** $\delta$-$k$-LWL distinguishes $G$ and $H$.

*Proof.* The proof idea is to show that $\delta$-$k$-LWL algorithm is powerful enough to detect distance-two-cliques of size $(k+1)$, which ensures the distinguishability of $G$ and $H$. Indeed, consider the $k$-tuple $P = ((1, \emptyset), (2, \emptyset), \ldots, (k, \emptyset))$ in $V(G)^k$. We claim that there is no tuple $Q$ in $V(H)^k$ such that the unrolling of $P$ is isomorphic to the unrolling of $Q$. Indeed, for the sake of contradiction, assume that there does exist $Q$ in $V(H)^k$ such that the unrolling of $Q$ is isomorphic to the unrolling of $P$. Comparing isomorphism types, we know that the tuple $Q$ must be of the form $((1, S_1), \ldots, (k, S_k))$.

Consider the depth-two unrolling of $P$: from the root vertex $P$, we can go down via two local-edges labeled 1, to hit the tuple $P_2 = ((2, \emptyset), (2, \emptyset), \ldots, (k, \emptyset))$. If we consider the depth-two unrolling of $Q$, the isomorphism type of $P_2$ implies that the vertices $(1, S_1)$ and $(2, S_2)$ must be at distance-two in the graph $H$. Repeating this argument, we obtain that $(1, S_1), \ldots, (k, S_k)$ form a distance-two-clique in $H$ of size $k$. Our goal is to produce a distance-two-clique in $H$ of size $k$, for the sake of contradiction.

For that, consider the depth-four unrolling of $P$: from the root vertex $P$, we can go down via two local-edges labeled 1 to hit the tuple $R = ((0, \emptyset), (2, \emptyset), \ldots (k, \emptyset))$. For each $2 \leq j \leq k$, we can further go down from $R$ via two local edges labeled $j$ to reach a tuple whose $1^{\text{st}}$ and $j^{\text{th}}$ entry is $(0, \emptyset)$. Similarly, for the unrolling of $Q$, there exists a subset $S_0 \subseteq E(0)$ and a corresponding tuple $R' = ((0, S_0), (2, S_2), \ldots, (k, S_k))$, such that for each $2 \leq j \leq k$, we can further go down from $R'$ via two local edges labeled $j$ to reach a tuple whose $1^{\text{st}}$ and $j^{\text{th}}$ entry is $(0, S_0)$. Comparing the isomorphism types of all these tuples, we deduce that $(0, S_0)$ must be at distance two from each of $(i, S_i)$ for $i \in [k]$. This implies that the vertex set $\{(0, S_0), (1, S_1), \ldots, (k, S_k)\}$ is a distance-two-clique of size $k + 1$ in $H$, which is impossible. Hence, there does not exist any $k$-tuple $Q$ in $V(H)^k$ such that the unrolling of $P$ and the unrolling of $Q$ are isomorphic. Hence, the $\delta$-$k$-LWL distinguishes $G$ and $H$. $\qquad\square$

Finally, we note that CFI graphs are standard tools from graph isomorphism theory, and are often used to analyze the power and limitations of WL-type algorithms. It follows from results of [15] that for every $k \geq 0$, $k$-WL fails to distinguish the graphs $G_k$ and $H_k$ of our constructed family. This finishes the proof of the proposition.

## D  Local $\delta$-$k$-dimensional Weisfeiler-Leman algorithm (Expanded)

In this section, we define the new *local $\delta$-$k$-dimensional Weisfeiler-Leman algorithm* ($\delta$-$k$-LWL). This variant of $\delta$-$k$-WL considers only local neighbors during the neighborhood aggregation process, and discards any information about the global neighbors. Formally, the $\delta$-$k$-LWL algorithm refines a coloring $C_i$ (obtained after $i$ rounds) via the aggregation map,

$$M_i^\delta(\mathbf{v}) = \left( \{\!\{ C_i(\phi_1(\mathbf{v}, w)) \mid w \in N(v_1) \}\!\}, \ldots, \{\!\{ C_i(\phi_k(\mathbf{v}, w)) \mid w \in N(v_k) \}\!\} \right), \qquad (3)$$

instead of Equation (8). That is, the algorithm only considers the local $j$-neighbors of the vertex $\mathbf{v}$ in each iteration. Therefore, the indicator function $\mathrm{adj}$ used in Equation (8) is trivially equal to $L$ here,

and is hence omitted. The coloring function for the $\delta$-$k$-LWL is defined by

$$C_{i+1}^{k,\delta}(\mathbf{v}) = (C_i^{k,\delta}(\mathbf{v}), M_i^{\delta}(\mathbf{v})).$$

We also define $\delta$-$k$-LWL$^+$, a minor variation of $\delta$-$k$-LWL. Later, we will show that $\delta$-$k$-LWL$^+$ is equivalent in power to $\delta$-$k$-WL (Theorem 10). Formally, the $\delta$-$k$-LWL$^+$ algorithm refines a coloring $C_i$ (obtained after $i$ rounds) via the aggregation function,

$$M^{\delta,+}(\mathbf{v}) = \big(\{\!\!\{(C_i(\phi_1(\mathbf{v},w)), \#_i^1(\mathbf{v},\phi_1(\mathbf{v},w))) \mid w \in N(v_1)\}\!\!\}, \ldots,$$
$$\{\!\!\{(C_i(\phi_k(\mathbf{v},w)), \#_i^k(\mathbf{v},\phi_k(\mathbf{v},w))) \mid w \in N(v_k)\}\!\!\}\big), \tag{4}$$

instead of $\delta$-$k$-LWL aggregation defined in Equation (9). Here, the function

$$\#_i^j(\mathbf{v},\mathbf{x}) = \big|\{\mathbf{w}\colon \mathbf{w} \sim_j \mathbf{v},\, C_i(\mathbf{w}) = C_i(\mathbf{x})\}\big|,$$

where $\mathbf{w} \sim_j \mathbf{v}$ denotes that $\mathbf{w}$ is $j$-neighbor of $\mathbf{v}$, for $j$ in $[k]$. Essentially, $\#_i^j(\mathbf{v},\mathbf{x})$ counts the number of $j$-neighbors (local or global) of $\mathbf{v}$ which have the same color as $\mathbf{x}$ under the coloring $C_i$ (i.e., after $i$ rounds). For a fixed $\mathbf{v}$, the function $\#_i^j(\mathbf{v},\cdot)$ is uniform over the set $S \cap N_j$, where $S$ is a color class obtained after $i$ iterations of the $\delta$-$k$-LWL$^+$ and $N_j$ denotes the set of $j$-neighbors of $\mathbf{v}$. Note that after the stable partition has been reached $\#_i^j(\mathbf{v})$ will not change anymore. Observe that each iteration of the $\delta$-$k$-LWL$^+$ has the same asymptotic running time as an iteration of the $\delta$-$k$-LWL.

The following theorem shows that the local variant $\delta$-$k$-LWL$^+$ is at least as powerful as the $\delta$-$k$-WL when restricted to the class of connected graphs. In other words, given two *connected* graphs $G$ and $H$, if these graphs are distinguished by $\delta$-$k$-WL, then they must also be distinguished by the $\delta$-$k$-LWL$^+$. On the other hand, it is important to note that, in general, the $\delta$-$k$-LWL$^+$ might need a larger number of iterations to distinguish two graphs, as compared to $\delta$-$k$-WL. However, this leads to advantages in a machine learning setting, see Section 6.

**Theorem 4** (restated, Theorem 2 in the main text). *For the class of connected graphs, the following holds for all $k \geq 1$:*

$$\delta\text{-}k\text{-LWL}^+ \equiv \delta\text{-}k\text{-WL}.$$

Along with Proposition 1, we obtain the following corollary relating the power of $k$-WL and $\delta$-$k$-LWL$^+$.

**Corollary 5** (restated, Corollary 3 in the main text). *For the class of connected graphs, the following holds for all $k \geq 2$:*

$$\delta\text{-}k\text{-LWL}^+ \sqsubset k\text{-WL}.$$

In fact, the proof of Proposition 1 shows that the infinite family of graphs $G_k, H_k$ witnessing the strictness condition can even be distinguished by $\delta$-$k$-LWL, for each corresponding $k \geq 2$. We note here that the restriction to connected graphs can easily be circumvented by adding a specially marked vertex, which is connected to every other vertex in the graph.

## D.1 Kernels based on vertex refinement algorithms

The idea for a kernel based on the $\delta$-$k$-LWL (and the other vertex refinements algorithms) is to compute it for $h \geq 0$ iterations resulting in a coloring function $C^{k,\delta}\colon V(G) \to \Sigma_i$ for each iteration $i$. Now, after each iteration, we compute a *feature vector* $\phi_i(G)$ in $\mathbb{R}^{|\Sigma_i|}$ for each graph $G$. Each component $\phi_i(G)_c$ counts the number of occurrences of $k$-tuples labeled by $c$ in $\Sigma_i$. The overall feature vector $\phi_{\mathrm{LWL}}(G)$ is defined as the concatenation of the feature vectors of all $h$ iterations, i.e., $\phi_{\mathrm{LWL}}(G) = \big[\phi_0(G), \ldots, \phi_h(G)\big]$. The corresponding kernel for $h$ iterations then is computed as $k_{\mathrm{LWL}}(G,H) = \langle \phi_{\mathrm{LWL}}(G), \phi_{\mathrm{LWL}}(H) \rangle$, where $\langle \cdot, \cdot \rangle$ denotes the standard inner product.

## D.2 Local converges to global: Proof of Theorem 2

The main technique behind the proof is to encode the colors assigned by the $k$-WL (or its variants) as rooted directed trees, called *unrolling trees*. The exact construction of the unrolling tree depends on the aggregation map $M(\cdot)$ used by the $k$-WL variant under consideration. We illustrate this construction for the $k$-WL. For other variants such as the $\delta$-$k$-WL, $\delta$-$k$-LWL, and $\delta$-$k$-LWL$^+$, we will specify analogous constructions.

Figure 2: Unrolling at the tuple $(u, v, w)$ of depth one.

**Unrollings ("Rolling in the deep")** Given a graph $G$, a tuple $\mathbf{v}$ in $V(G)^k$, and an integer $\ell \geq 0$, the *unrolling* $\mathsf{UNR}\,[G, \mathbf{v}, \ell]$ is a rooted, directed tree with vertex and edge labels, defined recursively as follows.

- For $\ell = 0$, $\mathsf{UNR}\,[G, \mathbf{v}, 0]$ is defined to be a single vertex, labeled with the isomorphism type $\tau(\mathbf{v})$. This lone vertex is also the root vertex.
- For $\ell > 0$, $\mathsf{UNR}\,[G, \mathbf{v}, \ell]$ is defined as follows. First, introduce a root vertex $r$, labeled with the isomorphism type $\tau(\mathbf{v})$. Next, for each $j \in [k]$ and for each $j$-neighbor $\mathbf{w}$ of $\mathbf{v}$, append the rooted subtree $\mathsf{UNR}\,[G, \mathbf{w}, \ell - 1]$ below the root $r$. Moreover, the directed edge $e$ from $r$ to the root of $\mathsf{UNR}\,[G, \mathbf{w}, \ell - 1]$ is labeled $j$ iff $\mathbf{w}$ is a $j$-neighbor of $\mathbf{v}$.

We refer to $\mathsf{UNR}\,[G, \mathbf{v}, \ell]$ as the unrolling of the graph $G$ *at* $\mathbf{v}$ *of depth* $\ell$. Figure 4 partially illustrates the recursive construction of unrolling trees: it describes the unrolling tree for the graph in Figure 3 at the tuple $(u, v, w)$, of depth 1. Each node $w$ in the unrolling tree is associated with some $k$-tuple $\mathbf{w}$, indicated alongside the node in the figure. We call $\mathbf{w}$ the tuple corresponding to the node $w$.

Analogously, we can define unrolling trees $\delta\text{-}\mathsf{UNR}$, $\mathsf{L\text{-}UNR}$, and $\mathsf{L^+\text{-}UNR}$ for the $k\text{-}\mathsf{WL}$-variants $\delta\text{-}k\text{-}\mathsf{WL}$, $\delta\text{-}k\text{-}\mathsf{LWL}$, and $\delta\text{-}k\text{-}\mathsf{LWL}^+$ respectively. The minor differences lie in the recursive step above, since the unrolling construction needs to faithfully represent the aggregation process.

- For $\delta\text{-}\mathsf{UNR}$, we additionally label the directed edge $e$ with $(j, L)$ or $(j, G)$ instead of just $j$, depending on whether the neighborhood is local or global.
- For $\mathsf{L\text{-}UNR}$, we consider only the subtrees $\mathsf{L\text{-}UNR}\,[G, \mathbf{w}, \ell - 1]$ for local $j$-neighbors $\mathbf{w}$.
- For $\mathsf{L^+\text{-}UNR}$, we again consider only the subtrees $\mathsf{L^+\text{-}UNR}\,[G, \mathbf{w}, \ell - 1]$ for local $j$-neighbors $\mathbf{w}$. However, the directed edge $e$ to this subtree is also labeled with the $\#$ counter value $\#_{\ell-1}^j(\mathbf{v}, \mathbf{w})$.

**Encoding colors as trees** The following Lemma shows that the computation of the $k\text{-}\mathsf{WL}$ can be faithfully encoded by the unrolling trees. Formally, let $\mathbf{s}$ and $\mathbf{t}$ be two $k$-vertex-tuples in $V(G)^k$.

**Lemma 6.** The colors of $\mathbf{s}$ and $\mathbf{t}$ after $\ell$ rounds of $k\text{-}\mathsf{WL}$ are identical if and only if the unrolling tree $\mathsf{UNR}\,[G, \mathbf{s}, \ell]$ is isomorphic to the unrolling tree $\mathsf{UNR}\,[G, \mathbf{t}, \ell]$.

*Proof.* By induction on $\ell$. For the base case $\ell = 0$, observe that the initial colors of $\mathbf{s}$ and $\mathbf{t}$ are equal to the respective isomorphism types $\tau(\mathbf{s})$ and $\tau(\mathbf{t})$. On the other hand, the vertex labels for the single-vertex graphs $\mathsf{UNR}\,[G, \mathbf{s}, 0]$ and $\mathsf{UNR}\,[G, \mathbf{t}, 0]$ are also the respective isomorphism types $\tau(\mathbf{s})$ and $\tau(\mathbf{t})$. Hence, the statement holds for $\ell = 0$.

For the inductive case, we proceed with the forward direction. Suppose that $k\text{-}\mathsf{WL}$ assigns the same color to $\mathbf{s}$ and $\mathbf{t}$ after $\ell$ rounds. For each $j$ in $[k]$, the $j$-neighbors of $\mathbf{s}$ form a partition $\mathbf{C}_1, \ldots, \mathbf{C}_p$ corresponding to their colors after $\ell - 1$ rounds of $k\text{-}\mathsf{WL}$. Similarly, the $j$-neighbors of $\mathbf{t}$ form a partition $\mathbf{D}_1, \ldots, \mathbf{D}_p$ corresponding to their colors after $\ell - 1$ rounds of $k\text{-}\mathsf{WL}$, where for $i$ in $[p]$, $\mathbf{C}_i$ and $\mathbf{D}_i$ have the same size and correspond to the same color. By inductive hypothesis, the corresponding depth $\ell - 1$ unrollings $\mathsf{UNR}\,[G, \mathbf{c}, \ell - 1]$ and $\mathsf{UNR}\,[G, \mathbf{d}, \ell - 1]$ are isomorphic, for every $\mathbf{c}$ in $\mathbf{C}_i$ and $\mathbf{d}$ in $\mathbf{D}_i$. Since we have a bijective correspondence between the depth $\ell - 1$ unrollings of the $j$-neighbors of

Figure 3: Unrollings $L_1 = \mathsf{L}^+\text{-}\mathsf{UNR}\,[G, \mathbf{s}, q]$ and $L_2 = \mathsf{L}^+\text{-}\mathsf{UNR}\,[G, \mathbf{t}, q]$ of sufficiently large depth.

$\mathbf{s}$ and $\mathbf{t}$, respectively, there exists an isomorphism between $\mathsf{UNR}\,[G, \mathbf{s}, \ell]$ and $\mathsf{UNR}\,[G, \mathbf{t}, \ell]$. Moreover, this isomorphism preserves vertex labels (corresponding to isomorphism types) and edges labels (corresponding to $j$-neighbors).

For the backward direction, suppose that $\mathsf{UNR}\,[G, \mathbf{s}, \ell]$ is isomorphic to $\mathsf{UNR}\,[G, \mathbf{t}, \ell]$. Then, we have a bijective correspondence between the depth $\ell - 1$ unrollings of the $j$-neighbors of $\mathbf{s}$ and of $\mathbf{t}$, respectively. For each $j$ in $[k]$, the $j$-neighbors of $\mathbf{s}$ form a partition $\mathbf{C}_1, \ldots, \mathbf{C}_p$ corresponding to their unrolling trees after $\ell - 1$ rounds of $k\text{-}\mathsf{WL}$. Similarly, the $j$-neighbors of $\mathbf{t}$ form a partition $\mathbf{D}_1, \ldots, \mathbf{D}_p$ corresponding to their unrolling trees after $\ell - 1$ rounds of $k\text{-}\mathsf{WL}$, where for $i$ in $[p]$, $C_i$, and $D_i$ have the same size and correspond to the same isomorphism type of the unrolling tree. By induction hypothesis, the $j$-neighborhoods of $\mathbf{s}$ and $\mathbf{t}$ have an identical color profile after $\ell - 1$ rounds. Finally, since the depth $\ell - 1$ trees $\mathsf{UNR}\,[G, \mathbf{s}, \ell - 1]$ and $\mathsf{UNR}\,[G, \mathbf{t}, \ell - 1]$ are trivially isomorphic, the tuples $\mathbf{s}$ and $\mathbf{t}$ have the same color after $\ell - 1$ rounds. Therefore, $k\text{-}\mathsf{WL}$ must assign the same color to $\mathbf{s}$ and $\mathbf{t}$ after $\ell$ rounds. $\qquad\square$

Using identical arguments, we can state the analogue of Lemma 12 for the algorithms $\delta\text{-}k\text{-}\mathsf{WL}$, $\delta\text{-}k\text{-}\mathsf{LWL}$, $\delta\text{-}k\text{-}\mathsf{LWL}^+$, and their corresponding unrolling constructions $\delta\text{-}\mathsf{UNR}$, $\mathsf{L}\text{-}\mathsf{UNR}$ and $\mathsf{L}^+\text{-}\mathsf{UNR}$. The proof is identical and is hence omitted.

**Lemma 7.** The following statements hold.

1. The colors of $\mathbf{s}$ and $\mathbf{t}$ after $\ell$ rounds of $\delta\text{-}k\text{-}\mathsf{WL}$ are identical if and only if the unrolling tree $\delta\text{-}\mathsf{UNR}\,[G, \mathbf{s}, \ell]$ is isomorphic to the unrolling tree $\delta\text{-}\mathsf{UNR}\,[G, \mathbf{t}, \ell]$.
2. The colors of $\mathbf{s}$ and $\mathbf{t}$ after $\ell$ rounds of $\delta\text{-}k\text{-}\mathsf{LWL}$ are identical if and only if the unrolling tree $\mathsf{L}\text{-}\mathsf{UNR}\,[G, \mathbf{s}, \ell]$ is isomorphic to the unrolling tree $\mathsf{L}\text{-}\mathsf{UNR}\,[G, \mathbf{t}, \ell]$.
3. The colors of $\mathbf{s}$ and $\mathbf{t}$ after $\ell$ rounds of $\delta\text{-}k\text{-}\mathsf{LWL}^+$ are identical if and only if the unrolling tree $\mathsf{L}^+\text{-}\mathsf{UNR}\,[G, \mathbf{s}, \ell]$ is isomorphic to the unrolling tree $\mathsf{L}^+\text{-}\mathsf{UNR}\,[G, \mathbf{t}, \ell]$.

**Equivalence** The following Lemma establishes that the local algorithm $\delta\text{-}k\text{-}\mathsf{LWL}^+$ is at least as powerful as the global $\delta\text{-}k\text{-}\mathsf{WL}$, for connected graphs, i.e., $\delta\text{-}k\text{-}\mathsf{LWL}^+ \sqsubseteq \delta\text{-}k\text{-}\mathsf{WL}$.

**Lemma 8.** Let $G$ be a connected graph, and let $\mathbf{s}$ and $\mathbf{t}$ in $V(G)^k$. If the stable colorings of $\mathbf{s}$ and $\mathbf{t}$ under the $\delta\text{-}k\text{-}\mathsf{LWL}^+$ are identical, then the stable colorings of $\mathbf{s}$ and $\mathbf{t}$ under $\delta\text{-}k\text{-}\mathsf{WL}$ are also identical.

*Proof.* Let $r^*$ denote the number of rounds needed to attain the stable coloring under $\delta\text{-}k\text{-}\mathsf{LWL}^+$. Consider unrollings $L_1 = \mathsf{L}^+\text{-}\mathsf{UNR}\,[G, \mathbf{s}, q]$ and $L_2 = \mathsf{L}^+\text{-}\mathsf{UNR}\,[G, \mathbf{t}, q]$ of sufficiently large depth $q = r^* + |V(G)| + 1$. Since $\mathbf{s}$ and $\mathbf{t}$ have the same stable coloring under $\delta\text{-}k\text{-}\mathsf{LWL}^+$, the trees $L_1$ and $L_2$ are isomorphic (by Lemma 13). Let $\theta$ be an isomorphism from $L_1$ to $L_2$.

We prove the following equivalent statement. If $L_1$ and $L_2$ are isomorphic, then for all $i \geq 0$, $\delta\text{-UNR}\,[G, \mathbf{s}, i] = \delta\text{-UNR}\,[G, \mathbf{t}, i]$. The proof is by induction on $i$. The base case $i = 0$ follows trivially by comparing the isomorphism types of $\mathbf{s}$ and $\mathbf{t}$.

For the inductive case, let $j \in [k]$. Let $\mathbf{X}_j$ be the set of $j$-neighbors of $\mathbf{s}$. Similarily, let $\mathbf{Y}_j$ be the set of $j$-neighbors of $\mathbf{t}$. Our goal is to construct, for every $j \in [k]$, a corresponding bijection $\sigma_j$ between $\mathbf{X}_j$ and $\mathbf{Y}_j$ satisfying the following conditions.

1. For all $\mathbf{x}$ in $\mathbf{X}_j$, $\mathbf{x}$ is a local $j$-neighbor of $\mathbf{s}$ if and only if $\sigma_j(\mathbf{x})$ is a local $j$-neighbor of $\mathbf{t}$.
2. For all $\mathbf{x}$ in $\mathbf{X}_j$, $\delta\text{-UNR}\,[G, \mathbf{x}, i-1] = \delta\text{-UNR}\,[G, \sigma_j(\mathbf{x}), i-1]$, i.e., $\mathbf{x}$ and $\sigma_j(\mathbf{x})$ are identically colored after $i-1$ rounds of $\delta$-$k$-$\mathsf{WL}$.

From the definition of $\delta\text{-UNR}$ trees, the existence of such $\sigma_1, \ldots, \sigma_k$ immediately implies the desired claim $\delta\text{-UNR}\,[G, \mathbf{s}, i] = \delta\text{-UNR}\,[G, \mathbf{t}, i]$. First, we show the following claim.

**Claim 9.** Let $\mathbf{C}$ be a color class in the stable coloring of $G$ under $\delta$-$k$-$\mathsf{LWL}^+$. Let $j \in [k]$. Then, $|\mathbf{C} \cap \mathbf{X}_j| = |\mathbf{C} \cap \mathbf{Y}_j|$.

*Proof.* Either $|\mathbf{C} \cap \mathbf{X}_j| = |\mathbf{C} \cap \mathbf{Y}_j| = 0$, in which case we are done. Otherwise, assume without loss of generality that $|\mathbf{C} \cap \mathbf{X}_j| \neq 0$. Let $\mathbf{x}$ in $\mathbf{C} \cap \mathbf{X}_j$. Since $G$ is connected, we can start from the root $s$ of $L_1$, go down along $j$-labeled edges, and reach a vertex $x$ such that $x$ corresponds to the tuple $\mathbf{x}$. Let $w$ be the parent of $x$, and let $\mathbf{w}$ be the tuple corresponding to $w$. Note that $\mathbf{x}$ is a local $j$-neighbor of $\mathbf{w}$. Moreover, the depth of $\mathbf{w}$ is at most $n-1$. Hence, the height of the subtree of $L_1$ rooted at $w$ is at least $q - (n-1) > r^*$.

Consider the tuple $\mathbf{z}$ corresponding to the vertex $z = \theta(w)$ in $L_2$. Observe that the path from the root $t$ of $L_2$ to the vertex $z = \theta(w)$ consists of $j$-labeled edges. Therefore, $\mathbf{z}$ is $j$-neighbor of $\mathbf{t}$, and hence $\mathbf{z}$ in $\mathbf{Y}_j$. The stable colorings of $\mathbf{w}$ and $\mathbf{z}$ under $\delta$-$k$-$\mathsf{LWL}^+$ are identical, because the subtrees rooted at $w$ and $z$ are of depth more than $r^*$. Let $\mathbf{C}$ denote the common color class of $\mathbf{w}$ and $\mathbf{z}$, in the stable coloring of $G$ under $\delta$-$k$-$\mathsf{LWL}^+$.

Since $\mathbf{x}$ is a local neighbor of $\mathbf{w}$, the agreement of the $\#$ function values ensures that the number of $j$-neighbors (local or global) of $\mathbf{w}$ in $\mathbf{C}$ is equal to the number of $j$-neighbors (local or global) of $\mathbf{z}$ in $\mathbf{C}$. Finally, the set of $j$-neighbors of $\mathbf{w}$ is equal to the set of $j$-neighbors of $\mathbf{s}$, which is $\mathbf{X}_j$. Similarily, the set of $j$-neighbors of $\mathbf{z}$ is equal to the set of $j$-neighbors of $\mathbf{t}$, which is $\mathbf{Y}_j$. Hence, $|\mathbf{C} \cap \mathbf{X}_j| = |\mathbf{C} \cap \mathbf{Y}_j|$. $\square$

Moreover, for each $j \in [k]$, the number of local $j$-neighbors of $\mathbf{s}$ in $\mathbf{C} \cap \mathbf{X}_j$ is equal to the number of local $j$-neighbors of $\mathbf{t}$ in $\mathbf{C} \cap \mathbf{Y}_j$. Otherwise, we could perform one more round of $\delta$-$k$-$\mathsf{LWL}^+$ and derive different colors for $\mathbf{s}$ and $\mathbf{t}$, a contradiction.

Hence, we can devise the required bijection $\sigma_j = \sigma_j^L \,\dot{\cup}\, \sigma_j^G$ as follows. We pick an arbitrary bijection $\sigma_j^L$ between the set of local $j$-neighbors of $\mathbf{s}$ inside $\mathbf{C}$ and the set of local $j$-neighbors of $\mathbf{t}$ inside $\mathbf{C}$. We also pick an arbitrary bijection $\sigma_j^G$ between the set of global $j$-neighbors of $\mathbf{s}$ inside $\mathbf{C}$ and the set of global $j$-neighbors of $\mathbf{t}$ inside $\mathbf{C}$. Clearly, $\sigma_j$ satisfies the first stipulated condition. By induction hypothesis, the second condition is also satisifed. Hence, we can obtain a desired bijection $\sigma_j$ satisfying the two stipulated conditions. Since we obtain the desired bijections $\sigma_1, \ldots, \sigma_k$, this finishes the proof of the lemma. $\square$

Finally, since for a graph $G = (V, E)$, $M_i^{\delta, \overline{\delta}}(\mathbf{v}) = M_i^{\delta, \overline{\delta}}(\mathbf{w})$ implies $M_i^{\delta, +}(\mathbf{v}) = M_i^{\delta, +}(\mathbf{w})$ for all $\mathbf{v}$ and $\mathbf{w}$ in $V(G)^k$ and $i \geq 0$, it holds that $\delta$-$k$-$\mathsf{WL} \sqsubseteq \delta$-$k$-$\mathsf{LWL}^+$. Together with Lemma 14 above, this finishes the proof of Theorem 10.

## E   Details on experiments and additional results

Here we give details on the experimental study of Section 6.

| Dataset | Properties | | | | | |
| --- | --- | --- | --- | --- | --- | --- |
| | Number of graphs | Number of classes/targets | ∅ Number of vertices | ∅ Number of edges | Vertex labels | Edge labels |
| ENZYMES | 600 | 6 | 32.6 | 62.1 | ✓ | ✗ |
| IMDB-BINARY | 1 000 | 2 | 19.8 | 96.5 | ✗ | ✗ |
| IMDB-MULTI | 1 500 | 3 | 13.0 | 65.9 | ✗ | ✗ |
| NCI1 | 4 110 | 2 | 29.9 | 32.3 | ✓ | ✗ |
| NCI109 | 4 127 | 2 | 29.7 | 32.1 | ✓ | ✗ |
| PTC_FM | 349 | 2 | 14.1 | 14.5 | ✓ | ✗ |
| PROTEINS | 1 113 | 2 | 39.1 | 72.8 | ✓ | ✗ |
| REDDIT-BINARY | 2 000 | 2 | 429.6 | 497.8 | ✗ | ✗ |
| YEAST | 79 601 | 2 | 21.5 | 22.8 | ✓ | ✓ |
| YEASTH | 79 601 | 2 | 39.4 | 40.7 | ✓ | ✓ |
| UACC257 | 39 988 | 2 | 26.1 | 28.1 | ✓ | ✓ |
| UACC257H | 39 988 | 2 | 46.7 | 48.7 | ✓ | ✓ |
| OVCAR-8 | 40 516 | 2 | 26.1 | 28.1 | ✓ | ✓ |
| OVCAR-8H | 40 516 | 2 | 46.7 | 48.7 | ✓ | ✓ |
| ZINC | 249 456 | 12 | 23.1 | 24.9 | ✓ | ✓ |
| ALCHEMY | 202 579 | 12 | 10.1 | 10.4 | ✓ | ✓ |
| QM9 | 129 433 | 12 | 18.0 | 18.6 | ✓(13+3D)[†] | ✓(4) |

Table 1: Dataset statistics and properties, [†]—Continuous vertex labels following [45], the last three components encode 3D coordinates.

## E.1 Datasets, graph kernels, and neural architectures

In the following, we give an overview of employed datasets, (baselines) kernels, and (baseline) neural architectures.

**Datasets** To evaluate kernels, we use the following, well-known, small-scale ENZYMES [98, 13], IMDB-BINARY, IMDB-MULTI [119], NCI1, NCI109 [109], PTC_FM [53][2], PRO-TEINS [31, 13], and REDDIT-BINARY [119] datasets. To show that our kernels also scale to larger datasets, we additionally used the mid-scale YEAST, YEASTH, UACC257, UACC257H, OVCAR-8, OVCAR-8H [117][3] datasets. For the neural architectures we used the large-scale molecular regression datasets ZINC [34, 57] and ALCHEMY [21]. We opted for not using the 3D-coordinates of the ALCHEMY dataset to solely show the benefits of the (sparse) higher-order structures concerning graph structure and discrete labels. To further compare to the (hierarchical) $k$-GNN [83] and $k$-IGN [77], and show the benefits of our architecture in presence of continuous features, we used the QM9 [91, 112] regression dataset.[4] To study data efficiency, we also used smaller subsets of the ZINC and ALCHEMY dataset. That is, for the ZINC 10K (ZINK 50K) dataset, following [34], we sampled 10 000 (50 000) graphs from the training, and 1 000 (5 000) from the training and validation split, respectively. For ZINC 10K, we used the same splits as provided by [34]. For the ALCHEMY 10K (ALCHEMY 50K) dataset, as there is no fixed split available for the full dataset[5], we sampled the (disjoint) training, validation, and test splits uniformly and at random from the full dataset. See Table 4 for dataset statistics and properties.[6]

**Kernels** We implemented the $\delta$-$k$-LWL, $\delta$-$k$-LWL$^+$, $\delta$-$k$-WL, and $k$-WL kernel for $k$ in $\{2, 3\}$. We compare our kernels to the Weisfeiler-Leman subtree kernel (1-WL) [100], the Weisfeiler-Leman Optimal Assignment kernel (WLOA) [68], the graphlet kernel [99] (GR), and the shortest-path kernel [13] (SP). All kernels were (re-)implemented in C++11. For the graphlet kernel we counted (labeled) connected subgraphs of size three.

**Neural architectures** We used the GIN and GIN-$\varepsilon$ architecture [115] as neural baselines. For data with (continuous) edge features, we used a 2-layer MLP to map them to the same number of components as the node features and combined them using summation (GINE and GINE-$\varepsilon$). For the evaluation of the neural architectures of Section 4, $\delta$-$k$-LGNN, $\delta$-$k$-

| Method | Dataset | | | | | |
|---|---|---|---|---|---|---|
| | YEAST | YEASTH | UACC257 | UACC257H | OVCAR-8 | OVCAR-8H |
| 1-WL | $88.8_{<0.1}$ | $88.8_{<0.1}$ | $96.8_{<0.1}$ | $96.9_{<0.1}$ | $96.1_{<0.1}$ | $96.2_{<0.1}$ |
| GINE | $88.3_{<0.1}$ | $88.3_{<0.1}$ | $95.9_{<0.1}$ | $95.9_{<0.1}$ | $94.9_{<0.1}$ | $94.9_{<0.1}$ |
| GINE-$\varepsilon$ | $88.3_{<0.1}$ | $88.3_{<0.1}$ | $95.9_{<0.1}$ | $95.9_{<0.1}$ | $94.9_{<0.1}$ | $94.9_{<0.1}$ |
| $\delta$-2-LWL | $89.2_{<0.1}$ | $88.9_{<0.1}$ | $97.0_{<0.1}$ | $96.9_{<0.1}$ | $96.4_{<0.1}$ | $96.3_{<0.1}$ |
| $\delta$-2-LWL$^+$ | $\mathbf{95.0}_{<0.1}$ | $\mathbf{95.7}_{<0.1}$ | $\mathbf{97.4}_{<0.1}$ | $\mathbf{98.1}_{<0.1}$ | $\mathbf{97.4}_{<0.1}$ | $\mathbf{97.7}_{<0.1}$ |

(The GINE, GINE-$\varepsilon$, $\delta$-2-LWL, and $\delta$-2-LWL$^+$ rows are grouped under "Local Neural".)

Table 2: Classification accuracies in percent and standard deviations on medium-scale datasets.

| | Set | Dataset | | | | | | | |
|---|---|---|---|---|---|---|---|---|---|
| | | ENZYMES | IMDB-BINARY | IMDB-MULTI | NCI1 | NCI109 | PTC_FM | PROTEINS | REDDIT-BINARY |
| $\delta$-2-WL | Train | 91.2 | 83.8 | 57.6 | 91.5 | 92.4 | 74.1 | 85.4 | – |
| | Test | 37.5 | 68.1 | 47.9 | 67.0 | 67.2 | 61.9 | 75.0 | – |
| $\delta$-2-LWL | Train | 98.8 | 83.5 | 59.9 | 98.6 | 99.1 | 84.0 | 84.5 | 92.0 |
| | Test | 56.6 | 73.3 | 50.2 | 84.7 | 84.2 | 60.3 | 75.1 | 89.7 |
| $\delta$-2-LWL$^+$ | Train | 99.5 | 95.1 | 86.5 | 95.8 | 94.4 | 96.1 | 90.9 | 96.2 |
| | Test | 52.9 | 75.7 | 62.5 | 91.4 | 89.3 | 62.6 | 79.3 | 91.1 |

Table 3: Training versus test accuracy of local and global kernels.

GNN, $k$-WL-GNN, we implemented them using PYTORCH GEOMETRIC [36], using a Python-wrapped C++11 preprocessing routine to compute the computational graphs for the higher-order GNNs. We used the GIN-$\varepsilon$ layer to express $f_{\mathrm{mrg}}^{W_1}$ and $f_{\mathrm{aggr}}^{W_2}$ of Section 4. Finally, we used the PYTORCH [90] implementations of the 3-IGN [77], and 1-2-GNN, 1-3-GNN, 1-2-3-GNN [83] made available by the respective authors.

For the QM9 dataset, we additionally used the MPNN architecture as a baseline, closely following the setup of [45]. For the GINE-$\varepsilon$ and the MPNN architecture, following Gilmer et al. [45], we used a complete graph, computed pairwise $\ell_2$ distances based on the 3D-coordinates, and concatenated them to the edge features. We note here that our intent is not the beat state-of-the-art, physical knowledge-incorporating architectures, e.g., DimeNet [66] or Cormorant [2], but to solely show the benefits of the (local) higher-order architectures compared to the corresponding (1-dimensional) GNN. For the $\delta$-2-GNN, to implement Equation (6), for each 2-tuple we concatenated the (two) node and edge features, computed pairwise $\ell_2$ distances based on the 3D-coordinates, and a one-hot encoding of the (labeled) isomorphism type. Finally, we used a 2-layer MLP to learn a joint, initial vectorial representation.

The source code of all methods and evaluation procedures is available at `https://www.github.com/chrsmrrs/sparsewl`.

## E.2 Experimental protocol and model configuration

In the following, we describe the experimental protocol and hyperparameter setup.

**Kernels** For the smaller datasets (first third of Table 4), for each kernel, we computed the (cosine) normalized gram matrix. We computed the classification accuracies using the $C$-SVM implementation of LIBSVM [20], using 10-fold cross-validation. We repeated each 10-fold cross-validation ten times with different random folds, and report average accuracies and standard deviations. For the larger datasets (second third of Table 4), we computed explicit feature vectors for each graph and used the linear $C$-SVM implementation of LIBLINEAR [35], again using 10-fold cross-validation (repeated ten times). Following the evaluation method proposed in [84], in the both cases, the $C$-parameter was selected from $\{10^{-3}, 10^{-2}, \dots, 10^2, 10^3\}$ using a validation set sampled uniformly at random from the training fold (using 10% of the training fold). Similarly, the number of iterations of the 1-WL, WLOA, $\delta$-$k$-LWL, $\delta$-$k$-LWL$^+$, and $k$-WL were selected from $\{0, \dots, 5\}$ using the validation set. Moreover, for the $\delta$-$k$-LWL$^+$, we only added the additional label function $\#$ on the last iteration to prevent overfitting. We report computation times for the 1-WL, WLOA, $\delta$-$k$-LWL, $\delta$-$k$-LWL$^+$, and $k$-WL with five refinement steps. All kernel experiments were conducted on

| Method | Dataset | | | | | |
|---|---|---|---|---|---|---|
| | ZINC (10k) | ZINC (50k) | ZINC (FULL) | ALCHEMY (10k) | ALCHEMY (50k) | ALCHEMY (FULL) |
| **Baseline** GINE-$\varepsilon$ | **0.278** ±0.022 | 0.145 ±0.006 | 0.084 ±0.004 | 0.185 ±0.007 -1.864 ±0.062 | 0.127 ±0.004 -2.415 ±0.053 | 0.103 ±0.001 -2.956 ±0.029 |
| 2-WL-GNN | 0.399 ±0.006 | 0.357 ±0.017 | 0.133 ±0.013 | 0.149 ±0.004 -2.609 ±0.029 | 0.105 ±0.001 -3.139 ±0.020 | 0.093 ±0.001 -3.394 ±0.035 |
| $\delta$-2-GNN | 0.374 ±0.022 | 0.150 ±0.064 | **0.042** ±0.003 | **0.118** ±0.001 -2.679 ±0.044 | **0.085** ±0.001 -3.239 ±0.023 | **0.080** ±0.001 -3.516 ±0.021 |
| $\delta$-2-LGNN | 0.306 ±0.044 | **0.100** ±0.005 | 0.045 ±0.006 | 0.122 ±0.003 -2.573 ±0.078 | 0.090 ±0.001 -3.176 ±0.020 | 0.083 ±0.001 -3.476 ±0.025 |

Table 4: Mean MAE (mean std. MAE, logMAE) on large-scale (multi-target) molecular regression tasks.

| Graph Kernel | Dataset | | | | | | | |
|---|---|---|---|---|---|---|---|---|
| | ENZYMES | IMDB-BINARY | IMDB-MULTI | NCI1 | NCI109 | PTC_FM | PROTEINS | REDDIT-BINARY |
| **Baseline** GR | <1 | <1 | <1 | 1 | 1 | <1 | <1 | 2 |
| SP | <1 | <1 | <1 | 2 | 2 | <1 | <1 | 1 035 |
| 1-WL | <1 | <1 | <1 | 2 | 2 | <1 | <1 | 2 |
| WLOA | <1 | <1 | <1 | 14 | 14 | <1 | 1 | 15 |
| **Global** 2-WL | 302 | 89 | 44 | 1 422 | 1 445 | 11 | 14 755 | OOM |
| 3-WL | 74 712 | 18 180 | 5 346 | OOT | OOT | 5 346 | OOM | OOM |
| $\delta$-2-WL | 294 | 89 | 44 | 1 469 | 1 459 | 11 | 14 620 | OOM |
| $\delta$-3-WL | 64 486 | 17 464 | 5 321 | OOT | OOT | 1 119 | OOM | OOM |
| **Local** $\delta$-2-LWL | 29 | 25 | 20 | 101 | 102 | 1 | 240 | 59 378 |
| $\delta$-2-LWL$^+$ | 35 | 31 | 24 | 132 | 132 | 1 | 285 | 84 044 |
| $\delta$-3-LWL | 4 453 | 3 496 | 2 127 | 18 035 | 17 848 | 98 | OOM | OOM |
| $\delta$-3-LWL$^+$ | 4 973 | 3 748 | 2 275 | 20 644 | 20 410 | 105 | OOM | OOM |

Table 5: Overall computation times for the whole datasets in seconds (Number of iterations for 1-WL, 2-WL, 3-WL, $\delta$-2-WL, WLOA, $\delta$-3-WL, $\delta$-2-LWL, and $\delta$-3-LWL: 5), OOT— Computation did not finish within one day (24h), OOM— Out of memory.

a workstation with an Intel Xeon E5-2690v4 with 2.60GHz and 384GB of RAM running Ubuntu 16.04.6 LTS using a single core. Moreover, we used the GNU C++ Compiler 5.5.0 with the flag -O2.

**Neural architectures** For comparing to kernel approaches, see Tables 1 and 5, we used 10-fold cross-validation, and again used the approach outlined in [84]. The number of components of the (hidden) node features in $\{32, 64, 128\}$ and the number of layers in $\{1, 2, 3, 4, 5\}$ of the GIN (GINE) and GIN-$\varepsilon$ (GINE-$\varepsilon$) layer were again selected using a validation set sampled uniformly at random from the training fold (using 10% of the training fold). We used mean pooling to pool the learned node embeddings to a graph embedding and used a 2-layer MLP for the final classification, using a dropout layer with $p = 0.5$ after the first layer of the MLP. We repeated each 10-fold cross-validation ten times with different random folds, and report the average accuracies and standard deviations. Due to the different training methods, we do not provide computation times for the GNN baselines.

For the larger molecular regression tasks, ZINC and ALCHEMY, see Table 7, we closely followed the hyperparameters found in [34] and [21], respectively, for the GINE-$\varepsilon$ layers. That is, for ZINC, we used four GINE-$\varepsilon$ layers with a hidden dimension of 256 followed by batch norm and a 4-layer MLP for the joint regression of the twelve targets, after applying mean pooling. For ALCHEMY and QM9, we used six layers with 64 (hidden) node features and a set2seq layer [108] for graph-level pooling, followed by a 2-layer MLP for the joint regression of the twelve targets. We used exactly the same hyperparameters for the (local) $\delta$-2-LGNN, and the dense variants $\delta$-2-GNN and 2-WL-GNN.

For ZINC, we used the given train, validation split, test split, and report the MAE over the test set. For the ALCHEMY and QM9 datasets, we uniformly and at random sampled 80% of the graphs for training, and 10% for validation and testing, respectively. Moreover, following [21, 45], we normalized the targets of the training split to zero mean and unit variance. We used a single model to predict all targets. Following [66, Appendix C], we report mean standardized MAE and mean standardized logMAE. We repeated each experiment five times (with different random splits in case of ALCHEMY and QM9) and report average scores and standard deviations.

| Graph Kernel | Dataset | | | | | |
|---|---|---|---|---|---|---|
| | YEAST | YEASTH | UACC257 | UACC257H | OVCAR-8 | OVCAR-8H |
| 1-WL | 11 | 19 | 6 | 10 | 6 | 10 |
| Local $\delta$-2-LWL | 1 499 | 5 934 | 1 024 | 3 875 | 1 033 | 4 029 |
| Local $\delta$-2-LWL$^+$ | 2 627 | 7 563 | 1 299 | 4 676 | 1 344 | 4 895 |

Table 6: Overall computation times for the whole datasets in seconds on medium-scale datasets (Number of iterations for 1-WL, $\delta$-2-LWL, and $\delta$-3-LWL: 2).

To compare training and testing times between the $\delta$-2-LGNN, the dense variants the $\delta$-2-GNN and 2-WL-GNN, and the (1-dimensional) GINE-$\varepsilon$ layer, we trained all four models on ZINC (10K) and ALCHEMY (10K) to convergence, divided by the number of epochs, and calculated the ratio with regard to the average epoch computation time of the $\delta$-2-LGNN (average computation time of dense or baseline layer divided by average computation time of the $\delta$-2-LGNN). All neural experiments were conducted on a workstation with four Nvidia Tesla V100 GPU cards with 32GB of GPU memory running Oracle Linux Server 7.7.

## Footnotes

[1] We define the 1-WL in the next subsection.

[2] https://www.predictive-toxicology.org/ptc/

[3] https://sites.cs.ucsb.edu/~xyan/dataset.htm

[4] We opted for comparing on the QM9 dataset to ensure a fair comparison concerning hyperparameter selection.

[5] Note that the full dataset is different from the contest dataset, e.g., it does not provide normalized targets, see https://alchemy.tencent.com/.

[6] All datasets can be obtained from http://www.graphlearning.io.