[Reviews · NeurIPS 2020]

Review 1

Summary and Contributions: ------------------------------------------------------------------- Post Rebuttal: After reading the response from the authors, I am sufficiently satisfied with majority of my concerns being addressed and the author's promise of adding the discussions requested by the reviewers/ tasks on link, triadic prediction to the revised version. My assessment remains positive. ------------------------------------------------------------------- The authors present local variants of the delta k-WL (k-Weisfeiler Leman) algorithm to exploit the sparsity present in real world graphs. Furthermore, they provide theoretical guarantees along with empirical evidence (on kernel and neural models based on their local variants) to compare the theoretical expressiveness ( and improved accuracy on experiments, improved computational timing and scope for better generalization) with respect to k-WL based kernels, GNNs, etc. More specifically, the authors propose the k-delta-LWL+ algorithm (and provide empirical kernel/ neural architectures based on them with improved results) which offer expressiveness at least as powerful as the k-delta-WL (on connected un-directed graphs) and more powerful than the k-WL algorithm.

Strengths: There has been a growing interest in the community in Graph Representation Learning over the past few years. Recent theoretical advances have shown the limitations of current graph neural networks and kernels in terms of their expressiveness being limited to 1-WL/ inability to scale to large graphs even for k =2,3 in k-WL. In this work the authors provide a theoretically sound local variant of k-delta- WL, which is able to reap benefits in terms of computational complexity for sparse graphs and back them up with empirical results (and provide kernel/ neural architectures based on them for k=2,3 with improved results). The local variants are shown to be equally powerful for connected undirected graphs as the k-delta-WL and more powerful than the traditional k-WL.

Weaknesses: 1. The techniques requires an initial feature vector based on isomorphism type of the tuple of size 'k' - the authors would be advised to stress that since it requires isomorphism testing of induced subgraphs of size k for initial coloring, this is not practically possible for larger values of k (which is a problem of the k-WL algorithms as well). 2. Since the representations of tuples of size k are learnt, it would be nice to compare results on tasks based on subgraphs of size 'k' - can you showcase results on subgraph problems (since you show results on k=2,3 - can you add tasks on link prediction, triad prediction)? 3. The authors don't specify the aggregation scheme used to go from tuple representations to complete graph representation - this is important as the number of tuples increases exponentially with the value of 'k' - while n^2 and n^3 computations might still be possible, larger values of k appear computationally intractable. The authors would be advised to clarify if they use all possible tuples or estimate the complete graph representation with an unbiased estimator.

Correctness: 1. The tree unrolling procedure employed to prove convergence of delta-k-LWL+ to delta-k-WL in Theorem 2 appears to be limited to only undirected graphs - for directed graph it requires strong connectivity. 2. The proofs of the other theorems, propositions appear to be correct to the best of understanding.

Clarity: The main paper is easy to read and understand - the appendix section needs to be be restructured - please see the minor concerns in the suggestion tab below.

Relation to Prior Work: The related work section is quite exhaustive and adequately represents the prior work in this domain.

Reproducibility: Yes

Additional Feedback: Some Concerns: 1. Address concerns brought up in the weaknesses section as possible/ or highlight the concerns in the paper. Other Minor Concerns: 1. Please note that the proof section in the appendix is ill structured - for example Theorem 6 in the main paper is made Theorem 8 in the appendix. Similarly, there is some confusion in the last lines of proof of theorem 2 (saying theorem 8 is proved there). Also most of the links to the \crefs in the appendix linking to equations, tables, etc don't function as intended. 2. There appears to be a lot of redundancy between the main paper and section 2 of the appendix . 3. The proof of proposition 1 is at the absolute end - why?. Please restructure for ease of readability.


Review 2

Summary and Contributions: In this paper, the authors propose some variants of the k-dimensional Weisfeiler-Lehman algorithm (k-WL) where the neighborhood of a k-tuple is a subset of the neighborhood defined by k-WL. For sparse graphs, the proposed variants are more efficient than the standard k-WL algorithm. Moreover, one of the variants is shown to be more powerful than the k-WL in terms of distinguishing between non-isomorphic connected graphs. Based on these algorithms, the authors derive novel kernels for graphs and graph neural networks. The proposed learning algorithms are evaluated on graph classification and graph regression datasets, where in most cases, they outperform the baselines.

Strengths: - The paper is well written and easy to follow. - In my opinion, the originality of the paper is high since it is both technically rich and the empirical results are impressive. The problem of reducing the complexity of the k-WL algorithm has not been studied in the past in this context. - Some variants of the proposed algorithm (e.g., the \delta-2-LWL^+ kernel) achieve very high performance. The 8 considered graph classification datasets are very well-studied, but still, some of the proposed kernels outperform state-of-the-art kernels/gnns by wide margins. - Overall, the paper seems to be proposing a novel contribution.

Weaknesses: - My main concern about this paper is the significance of the theoretical results. For instance, the authors show that the \delta-2-LWL^+ and \delta-2-WL algorithms are equally powerful in terms of distinguishing non-isomorphic graphs. However, in Table 1, we can see that the \delta-2-LWL^+ kernel largely outperforms the \delta-2-WL kernel on almost all datasets. Therefore, it seems that there is a gap between theory and practice. The derived theoretical results apply to a different problem (graph isomorphism) than the ones considered in the paper. In my view, it is important to devote more space to explaining how the features extracted by \delta-2-LWL^+ kernel are different from these of \delta-2-WL, and why they lead to better performance. - It is not clear to me what is the intuition behind the \delta-k-LWL^+ variant of the proposed algorithm. I think that the authors should make clear how did they come up with this variant and what are the advantages of this variant compared to other potential variants. - Another limitation of the proposed algorithms (mentioned by the authors in the supplementary material) is their high computational and space complexity. For instance, all graph classification datasets except Reddit-Binary contain relatively small graphs. On Reddit-Binary, most of the proposed kernels ran out of memory, while it took more than 23 hours to \delta-2-LWL^+ kernel to compute the kernel matrix. Thus, it is not clear if the proposed algorithms are useful for practical applications. - In most graph classification/regression experiments, the authors compare their kernels/graph neural networks against a single graph neural network, i.e., GIN. It would be nice if the authors included more recently-proposed graph neural networks into their list of baselines.

Correctness: The claims, the method and the empirical methodology all appear to be correct.

Clarity: The paper is well-written and easy to read.

Relation to Prior Work: In the introduction, the authors discuss how this work differs from previous studies.

Reproducibility: Yes

Additional Feedback: - In the graph classification experiments, the reported results for GIN are lower than those in the original paper. I wonder if the authors used the code provided by the authors of that paper. I guess the difference is due to the experimental protocol, however, I would expect it not to be that large. - Throughout the paper, the authors refer to the studied test of isomorphism as "Weisfeiler-Leman". On the other hand, in the literature the test is known as "Weisfeiler-Lehman". Is this just a typo or is there some reason behind that? - The proof of Proposition 1 is not complete. The authors do not show that the k-WL algorithm cannot distinguish the constructed graphs (G_k and H_k). Typos ----- p.4, l.161: "in Equation (6)" => There is no Equation (6) in the main paper p.5, l.181: "see Figure 4" => I think the authors mean Figure 2 appendix, p.5, l.198: "UNR[G, s, \ell]" => "UNR[G, v, \ell]" appendix, p.5, l.201: "\tau(s)" => "\tau(v)" appendix, p.5, l.206: "Figure 4" => "Figure 2" appendix, p.5, l.207: "Figure 3" => "Figure 1" ============= After Author's Response =============== I thank the authors for their feedback to my concerns. I have increased my score by 1.


Review 3

Summary and Contributions: The paper introduces a new class of methods for supervised learning on graphs relying on the higher-dimensional Weisfeiler-Leman (WL) algorithm. To tackle known issues present in existing k-WL inspired approaches (namely overfitting and complexity), the paper suggests to use local variants. The work then supports its claims theoretically and validate empirically.

Strengths: The paper brings forward clear claims and back them with theoretical support, composed of both novel ideas and derivation obtained from the literature. The paper is well constructed and the theoretical framework used is clear (after a few reading passes). The main contribution is to prove that the expressive power of the \delta-k-LWL+ algorithm is higher than the existing algorithm k-WL and is supported by solid empirical results (the performance of the proposed approach).

Weaknesses: The submission explores the expressivity of local variations of the k-dimensional WL algorithms and roots its motivation in the limitations of classical k-WL methods such as difficulty to scale and overfitting. However, while the suggested approach showcases impressive classification performance in the benchmark data sets, it is unfair to say that (1) overfitting is prevented (Table 2) and that (2) the proposed methods are scalable (OOM issues for slightly bigger data sets). In particular, it would be interesting to know the effective runtimes instead of comparing speedups (in order to better evaluate practical use). Furthermore, general limitations of the method are omitted, e.g. issues related to the combinatorial explosion of k-tuples are only very briefly mentioned in the main paper (Appendix F does a better job). Overall, this contributes to the impression that no strong practical motivations pushed the development of the method and that the work is mainly motivated by theoretical interests (which is absolutely fine but should be owned more). On another hand, the contributions, while theoretically sound and interesting, seem to be of limited novelty as they extend the existing \delta-k-WL by only looking at local neighbours (in the case of \delta-k-LWL) and counting them (for the + variant). While this takes nothing from the fact that the theoretical results are well derived, it could indicate a limited relevance for the broader community at NeurIPS. The relation to neural approach is in that aspect perhaps more relevant to a broader audience, given the large interest for this class of models and authors should expand on that. Finally, while the paper is clear, it is not the most accessible for readers who are unfamiliar with the topic, some efforts in the direction of providing intuitive descriptions could be made. ---- I'd like to thank the authors of the paper for addressing some of the concerns raised here. This further validates my overall score.

Correctness: Yes, most claims are correct. I did not run the provided code.

Clarity: Yes. The abstract can be misleading at a first glance as it mentions the _local_ nature of 1-WL as a limiting factor and then proposes a _local_ variant of k-WL: it might be clearer to reformulate the limitation. Nonetheless, the paper in its raw conference format is almost not self-sufficient and a read through the appendix is needed. In case of publication, I would advise to slightly relax the page limit to make the main paper more readable. Some internal links seem to be broken (see below, perhaps due to the fact that Appendix and Main paper contain multiple instance of the same theorems/equations).

Relation to Prior Work: The paper clearly refers, explains, and exploits prior work. The differences with existing work are explained.

Reproducibility: Yes

Additional Feedback: Minor comments: L. 154, broken link to equation (7), probably meant equation (3) L. 181 Figure 4 in appendix is not the correct pointer L. 349 App. Theorem 14?


Review 4

Summary and Contributions: Summary: 1) Problem: In the work, the authors are facing the problem of WL-based graph learning, e.g., distinguishing non-isomorphic graphs. 2) Keypoint: They found the local neighborhood feature could help the k-WL method to perform better theoretically and practically. 3) Solution: They propose a method called local \delta-k-dimensional WL algortihm wihch is a local version of the existing k-WL method. The merit of \delta-k-dimension LWL is considering a subset of the original neighborhood in each iteration. 4) Contributions: a) They theoretically analyze the strength of a variant of their local algorithm and prove that it is strictly more powerful in distinguishing non-isomorphic graphs compared to the original k-WL. b) They prove that the \delta-k-LWL is more powerful than the original k-WL by decising a hierachy of pairs of non-isomorhphic graphs. c) They propose a higher-order graph neral network architecture called \delta-k-LGNN, which is proven to have the same expressive power as the \delta-k-LWL. d) They also show that \delta-k-LWL architecture has better generalization abilities compared to other architectures based on the k-WL.

Strengths: As expected for a non-isomorphic graph distinguishable problem, the authors are supposed to show both theoretical improvement compared to existing WL related work. And they also need to support their claim with convincible empirical experiment design. 1) For the theoretical part, they provide a clear proof structure for the core method, \delta-k-LWL. Those claims of advantage of \delta-k-LWL are shown convincible with these proofs and clear logic. 2) The authors provide the discussion of the limitatio nor \delta-k-LWL. Then they show their higher-order neural architectures, \delta-k-LGNN as instead. And they also show that \delta-k-GNN inherits the main strength of \detla-k-LWL. 3) The empirical experiments are provided with different experiment settings and enough types of benchmark datasets for the graph distinguishing purpose. They support the main claim of \delta-k-LWL method and make it convincible. Compared to the previous works, the authors utilize the proof and experiments to show the important of local neighborhood. Such insight could draw more attention to the graph leanring problem and it could also help various research area to explore the possiblity of their own. In my point of view, the work is highly relevant to the NeurIPS community, which both show the improvment of the graph learning algorithm in the theoretical view and also provide the perspective of graph learning in practice.

Weaknesses: The overall of the paper is well written to present the key idea of \delta-k-LWL. Some parts can still be improved to make the submission better. 1) Some notaions missing: In the main document, there are some notations found missing, but they are found in the supplementary materials. For the readers who want to check the details of no matter the claims or the proofs, it could cause distraction and confusion. For example, in p3line131, the "L or G" is presented to show the Local or Global. However, the such definition is not presented before it was used. Although the reader can get the notations in the supplementary, it will be more reader-friendly to predefine these before line 131. 2) Experiment explanation: the overall experiment is fine. The authors make many comparison to show how powerful the \detal-k-LWL is. But there are some OOT and OOM shown in the Table 1 p7. In the experiment explanation, the authors just state the comparison results and the "OOT OOM" happened, but without proper explanation what actually happened in the implementation. It makes the reader worries about the generalization of this main method since the computation machine is powerful and the memory size is quite large based on the claim of the author in the supplementary. 3) The conclusion of \delta-k-LGNN seems not as impressive as the \delta-k-LWL in the experiment part. Does that mean \delta-k-LGNN has some limitation in the real-world application?

Correctness: The overall looks convincible, but the authors need to present more explanation on the empirical part of the \delta-k-LGNN which doesn't provide enough contents to support the outperformance of \delta-k-LGNN.

Clarity: Yes, it is.

Relation to Prior Work: Yes, it is. The authors take efforts to discuss the related work in the supplementary.

Reproducibility: Yes

Additional Feedback: 1) Typo: p2.line.52 "that is has the same expressive power" => "that has the same experessive power" 2) Answers in experiment part: Since the authors raise questions at the beginning of section 5, it could be better that the author draw the conclusion to the question with just one or two sentences before starting the discussion of the details. Especially, some questions can be answered by yes or no.

[Author Response · NeurIPS 2020]

*We thank all reviewers for their fair and constructive reviews.* 👍

**General remarks** We will revise and restructure the appendix (and the main paper), especially concerning refer-
ences/theorem numbering and redundancy between the main text.

*Scalability* We agree that our methods are not ready for industrial-scale graphs. We view our contribution as method-
ological work that is the first step to make higher-order methods more practical by leveraging *sparsity*, which is perhaps
the most significant parameter associated with a class of graphs. Moreover, we want to stress that we empirically
verified that our method offers significant benefits (compared to standard GNNs) in the regime of small graphs such as
molecules. We will integrate the discussion of Appendix F into the main text (complexity, sampling, . . . ).

**R1** *Computation of isomorphism type* We agree that the method is only of theoretical interest for a large choice of $k$.
Nonetheless, we believe that we provided sufficient empirical evidence showing that for small $k \in \{2, 3\}$, the method
provides benefits over standard GNN and other higher-order architectures. Note that strictly speaking, the $k$-WL does
not need to perform an isomorphism test. It merely performs an equality test. Lines 97-98 (in the "appendix.pdf")
describe the condition for two tuples to have the same initial labeling: it is only the identity mapping $\mathsf{id} \colon [k] \mapsto [k]$
which we demand to be a (partial) graph isomorphism. Hence, we only need to check "equality" of the two tuples and
not their "isomorphism". If we had considered two *sets* of $k$ vertices instead of two *tuples* of $k$ vertices, then we would
have needed isomorphism testing (as the reviewer suggests).
*Subgraph learning* Yes, we agree that results on link and triadic prediction would further highlight the approach's
generality. We plan to include them in a revised version/future work and add a discussion to the main paper.
*Aggregation scheme* We considered all tuples. We will make the complexity, especially the dependence on $k$, clearer in
the main paper, e.g., by including the discussion of Appendix F in the main paper.
*Directed graphs* Thank you for bringing this to our attention. The applicability of our arguments to the direct case
depends on the way we define the $k$-WL for directed graphs. It is typical to consider both in-neighbours and out-
neighbours, but separately. Hence for a tuple T, we have "local-in"-neighbors and "local-out"-neighbors. The local
unfolding at the tuple T will have both in-neighbours and out-neighbours at depth-1, and so on. Therefore, in this kind
of WL, we are essentially dealing with the underlying undirected graph of the given directed graph. If the graph is
weakly connected, we will still see all 1-neighbours of a tuple (local-in,local-out, global) somewhere down the local
unfolding tree. In that case, we are fine with weak connectivity. Of course, we could choose to define a WL-variant
with only "local-out"-neighbors, where we agree that strong connectivity will be necessary. Since the typical notion of
sparsity for directed/undirected graphs is the same, i.e., low edge-density, we would argue that it is natural to stick with
the first formalism. In that case, weak connectivity will suffice. We will clarify this in the revised version.

**R2** *Performance of $\delta$-2-LWL$^+$* The $\delta$-2-LWL$^+$ is as a middle ground between the purely local $\delta$-2-LWL and the global
$\delta$-2-WL. The +-Version achieves good generalization performance due to slower convergence while preserving certain
global information, which is needed to derive Theorem 2. We will expand on this in the revised version.
*Other GNN architectures* We agree and will include more recently-proposed GNNs as baselines.
*Lower results for GIN* We used the implementation available in Pytorch Geometric. The lower numbers are because
we did not use an initial one-hot degree feature for datasets that did not provide node labels. We will add a comment
and add a row for results using degree features. (Note that all kernels are computed without this information)
*Leman vs. Lehman* Leman/Lehman personally stated that he prefers the transcription Leman (through private commu-
nication with Russian graph theorist Ilia Ponomarenko).[1] Moreover, the spelling Leman is also used throughout the
theory community.
*Proof of Prop. 1* Thank you for bringing this to our attention. The missing part directly follows from the CFI
construction outlined in [15]. However, a formal proof is quite involved and would require repeating the reasoning of
the paper above. We will add a proof sketch to the revised version of the paper.

**R3** *Scalability/limitations* See general remarks above. We will report absolute running times (<1h for the local
architecture on the 10k subset.).
*Readability/relevance to a broader audience* We will incorporate intuition in the revised paper and emphasize the
neural architecture. Moreover, we will try to make the main paper as self-contained as possible.

**R4** *Notation* We agree and will make the main text more self-contained.
*$\delta$-k-LGNN* We note that the $\delta$-k-LGNN is slightly weaker than the $\delta$-k-GNN as it does not use the $\#$ labeling. (It
could be included but would require preprocessing.). Moreover, observe that the $\delta$-k-GNN may learn to combine the
local and global neighborhood information in a more fine-grained way compared to the kernel (due to its discrete
nature). Moreover, we stress here that the $\delta$-k-LGNN still achieves good performance while being much faster than the
$\delta$-$k$-GNN or $k$-WL-GNN. We will add an expanded discussion in the revised paper.
*Explanation on experiment part* Thank you for the good suggestion; we will incorporate it into a revised version.

## Footnotes

[1]https://www.iti.zcu.cz/wl2018/pdf/leman.pdf


[Meta-Review · NeurIPS 2020]

The authors present a local variant of k-WL-equivalent provably expressive graph neural networks. - the reviewers appreciated the importance and timeliness of the topic, in particular the theoretical expressive power of graph neural networks - the paper proposes a novel way of reducing the complexity of k-WL - impressive experimental results The rebuttal was read and discussed by the reviewers. The rebuttal addressed most concerns raised, yet, some concern still remained about the gap between theory and practice. Overall the reviewers are positive and our recommendation is to accept the paper.